



**A Model-Independent Data Assimilation (MIDA) module and its applications in ecology**
Xin Huang[1,2], Dan Lu[3], Daniel M. Ricciuto[4], Paul J. Hanson[4], Andrew D. Richardson[1,2], Xuehe
Lu[5], Ensheng Weng[6,7], Sheng Nie[8], Lifen Jiang[1], Enqing Hou[1], Igor F. Steinmacher[2], Yiqi
Luo[1,2,9]
Center for Ecosystem Science and Society, Northern Arizona University, Flagstaff, AZ, USA
School of informatics, Computing, and Cyber Systems, Northern Arizona University, Flagstaff,
AZ, USA
Computational Sciences and Engineering Division, Climate Change Science Institute,
Oak Ridge National Laboratory, Oak Ridge, TN, USA
Environmental Sciences Division, Climate Change Science Institute, Oak Ridge National
Laboratory, Oak Ridge, TN, USA
International Institute for Earth System Science, Nanjing University, Nanjing, China
Center for Climate Systems Research, Columbia University, New York, USA
NASA Goddard Institute for Space Studies, New York, USA
Key Laboratory of Digital Earth Science, Aerospace Information Research Institute, Chinese
Academy of Sciences, Beijing, China
Department of Biological Sciences, Northern Arizona University, Flagstaff, AZ, USA
*Correspondence to:* Xin Huang (xh59@nau.edu)





**ABSTRACT**
Models are an important tool to predict Earth system dynamics. An accurate prediction of future
states depends on not only model structures but also parameterizations. Model parameters can be
constrained by data assimilation. However, applications of data assimilation to ecology are
restricted by highly technical requirements such as model-dependent coding. To alleviate this
technical burden, we developed a model-independent data assimilation (MIDA) module. MIDA
works in three steps including data preparation, execution of data assimilation, and visualization.
The first step prepares prior ranges of parameter values, a defined number of iterations, and
directory paths to access files of observations and models. The execution step calibrates
parameter values to best fit the observations and estimates the parameter posterior distributions.
The final step automatically visualizes the calibration performance and posterior distributions.
MIDA is model independent and modelers can use MIDA for an accurate and efficient data
assimilation in a simple and interactive way without modification of their original models. We
applied MIDA to four types of ecological models: the data assimilation linked ecosystem carbon
(DALEC) model, a surrogate-based energy exascale earth system model the land component
(ELM), nine phenological models and a stand-alone biome ecological strategy simulator
(BiomeE). The applications indicate that MIDA can effectively solve data assimilation problems
for different ecological models. Additionally, the easy implementation and model-independent
feature of MIDA breaks the technical barrier of black-box applications of data-model fusion in
ecology. MIDA facilitates the assimilation of various observations into models for uncertainty
reduction in ecological modeling and forecasting.
**Keywords:**
Parameter uncertainty quantification, Data assimilation, Modules, Ecological models





## 1. Introduction

Ecological models require a large number of parameters to simulate biogeophysical and
biogeochemical processes (Bonan, 2019; Ciais et al., 2013; Friedlingstein et al., 2006), and
specify model behaviors (Luo et al., 2016; Luo and Schuur, 2020). Parameter values in
ecological models are mostly determined in some *ad hoc* fashions (Luo et al., 2001), leading to
considerable biases in predictions (Tao et al., 2020). The situation becomes even worse when
more detailed processes are incorporated into models (De Kauwe et al., 2017; Lawrence et al.,
2019). Data assimilation (DA), a statistically rigorous method to integrate observations and
models, is gaining increasing attention for parameter estimation and uncertainty evaluation. It
has been successfully applied to many ecological models (Fox et al., 2009; Keenan et al., 2012;
Richardson et al., 2010; Safta et al., 2015; Wang et al., 2009; Williams et al., 2005; Zobitz et al.,
2011). However, almost all those DA studies require model-dependent, invasive coding. This
requires a DA algorithm to be programmed for a specific model. Such model-dependent coding
creates a large technical barrier for ecologists to use DA to solve prediction and uncertainty
quantification problems in ecology. Thus a model-independent DA toolkit is required to facilitate
the use of DA technique in ecology.
DA is a powerful approach to combine models with observations and can be used to
improve ecological research in several ways (Luo et al., 2011). First, DA can be used for
parameter estimation (Bloom et al., 2016; Hararuk et al., 2015; Hou et al., 2019; Ise and
Moorcroft, 2006; Ma et al., 2017; Ricciuto et al., 2011; Scholze et al., 2007). It enables the
optimization of parameter values across sites, time and treatments (Li et al., 2018; Luo and
Schuur, 2020). For example, Hararuk and his colleagues applied DA to a global land model and
substantially improved the explanability of the global variation in soil organic carbon (SOC)



from 27% to 41% (Hararuk et al., 2014). When DA was combined with deep learning to improve
spatial distributions of estimated parameter values, for example, the Community Land Model
version 5 (CLM5) predicted the SOC distribution in the US continent with much higher $R^2$ of
0.62 than CLM5 with default parameters ($R^2 = 0.32$) (Tao et al., 2020). Second, DA can be used
to select alternative model structures to better represent ecological processes (Liang et al., 2018;
Van Oijen et al., 2011; Shi et al., 2018; Smith et al., 2013; Williams et al., 2009). DA was used
to evaluate four models and a two-pool interactive model was selected after DA to best represent
SOC decomposition with priming (Liang et al., 2018). Additionally, DA can be applied for data-
worth analysis to locate the most informative data to reduce uncertainty, thus guiding the sensor
network design. (Keenan et al., 2013; Raupach et al., 2005; Shi et al., 2018; Williams et al.,
2005). One DA study at Harvard Forest (Keenan et al., 2013) indicated that only a few data
sources contributed to the significant reduction in parameter uncertainty. Overall, DA is essential
for ecological modeling and forecasting (Jiang et al., 2018) and is helpful for evaluation of
different inversion methods (Fox et al., 2009).
Applications of traditional DA to ecological research require highly technical skills of
users. A successful DA application usually involves model-dependent coding to integrate
observations into models. This requires users to have knowledge about model programing. For
example, if a complex model (e.g., the community land model) is used in DA, users need to
know the programming language (e.g., Fortran) of the model and its internal content to write DA
algorithm into the model source code before DA can be conducted. The learning curve for model
programing is steep for general ecologists. Furthermore, users often need to update the
programming knowledge when a different model is used in DA. For example, scientists who
implemented the DA algorithm coded in MATLAB ( Xu et al., 2006) to an ecosystem carbon



cycle model programmed in Fortran (e.g., TECO) need to understand both MATLAB and
Fortran (Ma et al., 2017). Moreover, DA often involves reading observation files about a specific
study site. As a result, users usually have to update the codes of model-dependent DA to read
new observations from every new study site.

A number of tools have been developed to facilitate DA applications (Table 1) but many

of them are model dependent, such as the Carbon Cycle Data Assimilation Systems (CCDAS)
(Rayner et al., 2005; Scholze et al., 2007), the Carbon Data Model Framework (CARDAMOM)
(Bloom et al., 2016), and the Ecological Platform for Assimilating Data (EcoPAD) data
assimilation systems (Huang et al. 2019). These tools combine DA algorithms with a specific
model. For example, CCDAS specified the DA algorithm to the Biosphere Energy Transfer
Hydrology (BETHY) model (Rayner et al., 2005). The hardcoding feature of aforementioned
tools make them inflexible to be applied to different models.

There are some model independent DA tools that are not tailored to a specific model,

such as Data Assimilation Research Testbed (DART) (Anderson et al., 2009), the open Data
Assimilation library (openDA) (Ridler et al., 2014), the Parallel Data Assimilation Framework
(PDAF) (Nerger and Hiller, 2013) and Parameter Estimation & Uncertainty Analysis software
suit (PEST) (Doherty, 2004).

However, these model-independent tools suffer from some limitations for a general and

flexible DA application. For example, openDA requires users to code three functions to initialize
a Java class (Ridler et al., 2014) (Table 1). DART enables incorporating a new model through a
range of interfaces (Anderson et al., 2009). It has been successfully applied to atmospheric and
oceanic models with currently available interfaces (Anderson et al., 2009; Raeder et al., 2012)
and recently to the community land model (Fox et al. 2019). It is likely that users may need to



prepare new interfaces for new ecological models to use DART. DART and PDAF adopted the
Ensemble Kalman Filter (EnKF) method (Evensen, 2003), which may makes it difficult to obey
mass conservation for biogeochemical models. This is because the parameter values estimated by
EnKF change each time when new data sets are assimilated (Allen et al., 2003; Gao et al., 2011;
Trudinger et al., 2007). The disruptive changes in estimated parameter values usually do not
reflect reality of biogeochemical cycles in the real world. PEST utilizes Levenberg-Marquardt
method (Levenberg, 1944) which is a local optimization method for parameter estimation. If the
relationship between simulation outputs and parameters are highly nonlinear, which is common
in ecological models, this method may trap into a locally optimization solution (Doherty, 2004).

In this work, we developed a model-independent DA module (MIDA) to enable a general

and flexible application of DA in ecology. MIDA was designed as a highly modular tool,
independent of specific models, and friendly to users with limited programming skills and/or
technical knowledge of DA algorithms. Additionally, MIDA implemented advanced Markov
Chain Monte Carlo (MCMC) algorithms for DA analysis which can accurately quantify the
parameter uncertainty with informative posterior distribution. The anticipated user community in
this initial phase of MIDA development is the biogeochemical modelers who are looking for
appropriate parameter estimation methods. In the following Section 2, we first introduce the
development details of MIDA and its usage. In Section 3, we demonstrate the application of
MIDA to four different types of ecological models. In Section 4, we discuss the strengths and
weaknesses of MIDA in ecological modelling and lastly we give our concluding remarks in
Section 5.

**2.  Model-independent data assimilation (MIDA)**





### 2.1 DA algorithm


DA is a statistical algorithm to constrain parameter values and estimate their posterior density
distributions through assimilating observations into a model. This algorithm successively
generates a new set of parameter values and requires model run with these new parameter values.
Then the misfit between model simulation outputs and observations is calculated to determine
whether this new set of parameter values will be accepted or not. The previously accepted
parameter values help to generate new parameter values in the next iteration. Each iteration
incorporates a model-dependent data exchange to transfer parameter values, model outputs,
observations, etc. between DA algorithm and the model. Traditional DA requires implementing
these data exchanges through model-specific programming into model code. As a result, a DA
application inevitably involves intrusive modification of the original model.

### 2.2 An overview of MIDA


MIDA (https://github.com/Celeste-Huang/MIDA, last access: Feb 2021) is a module that allows
for automatic implementation of data assimilation without intrusive modification or coding of the
original model. Its workflow includes three steps: data preparation, data assimilation, and
visualization (Fig. 1). Step 1 (data preparation) is to establish the standardized data exchange
between DA algorithm and the model. Step 2 (data assimilation) is to run DA as a black box
independent of the model. Step 3 (visualization) is to diagnose parameter uncertainty after DA.
The modularity of the 3-step workflow is designed to enable MIDA for a rapid DA application
and adaption to a new model. In the following, we introduce the three-step workflows of MIDA,
its technical implementation and usage in detail.



### 2.3 Step 1: Data preparation

Step 1 is designed to initialize data exchange to transfer parameter values, model outputs, observations and their variances between DA algorithm and the model to be used. Four types of information are required either from interactive input or by modifying the 'namelist.txt' file (Fig. 1). The first type is about DA configuration, including the number of sampling series in DA and the working path where the outputs of DA will be saved. The number of a sampling series is essential in a DA task to define how many times parameter values are sampled to run the model. The second type of information is about parameter ranges and their covariance. The third is the model executable file. Finally, the fourth type is an output configuration file which contains the file paths of model outputs, observations, and their variance. This file also instructs how to read model outputs and compare each output with corresponding observation.

Traditional DA requires users to modify the code of model to incorporate the process of data exchange between DA algorithm and the model. Therefore, the program of data exchange in traditional DA is model-specific and users need to repeat such program when a new model comes. In MIDA, the process of data exchange calls a model executable file which hinders the details of model code. When applied to a new model, MIDA only requires users to provide a different model executable file in the 'namelist.txt' file and does not involve any additional coding in either the model or MIDA. Thus, MIDA lowers the technical barrier for general ecologists to conduct DA.

Traditional DA usually preset the number of parameters and the model outputs according to a specific model before initializing the data exchange. This is because data exchange between DA algorithm and model uses memory to transfers items such as parameter values. Instead, MIDA organize items in data exchange using different files. Items in data exchange are decided





by the data file loaded when MIDA is running. The number of parameter values, for example,
will be decided after the file of parameter range is read in MIDA. Through modifying files,
MIDA allows making efficient choices about the model-related items in data exchange. Thus,
MIDA is highly flexible and modular for DA with different models.

Traditional DA also preset observation types in the data exchange according to a specific

study before the data exchange. For example, if the traditional DA uses carbon flux observation,
it cannot switch to satellite remote sensing products without additional coding. MIDA uses the
concepts of object-orient programming (Mitchell and Apt, 2003) and dynamic initialization
(Cline et al., 1998) in computer science to provide a homogenous way to create various
observation types from a unified prototype class. A prototype class includes variables to store
observations and their variance and functions (e.g., read from observation files). The values in
variables are dynamically decided after the observation files are loaded when MIDA is running.
Different observation types derive from the prototype class with a high degree of reusability of
most functions. In such way, MIDA only requires users to provide different filenames of the
observations to be integrated in DA. Therefore, MIDA is highly flexible and modular for DA to
assimilate various observations.

**2.4 Step 2: Execution of data assimilation**
After the establishment of the standardized data exchange (step 1), step 2 is to run DA as a black
box for users without knowledge of DA itself. Notwithstanding the black-box goal, this section
provides a general description of DA below.

Data assimilation as a process integrates observations into a model to constrain

parameters and estimate parameter uncertainties. Data assimilation usually uses some types of





sampling algorithms, such as Markov chain Monte Carlo (MCMC), to generate posterior
parameter distribution under a Bayesian interference framework (Box and Tiao, 1992). This
version of MIDA uses MCMC algorithm implemented by the Metropolis-Hasting (MH)
sampling method (Harrio et al., 2001). The future version of MIDA could incorporate other data
assimilation algorithms. Each iteration in the Metropolis-Hasting sampling includes a proposing
phase and a moving phase. The proposing phase generates a new set of parameter values based
on the starting point for the first iteration or current accepted parameter values in the following
iterations. If parameter covariance ($cov_{param}$) is specified in step 1 on data preparation, this
proposing phase will draw new parameter values ($P_{new}$) within the prior ranges from a Gaussian
distribution $N(P_{old}, cov_{param})$ where $P_{old}$ is the predecessor set of parameter values. Without
parameter covariance, new set of parameter values will be generated from a uniform distribution
within the prior ranges.

The moving phase first calculates mismatches between observations and the model

simulation with the new set of parameter values as a cost function ($J_{new}$ in Eq.1) (Xu et al.

2006):

$$J_{new} = \sum_{i=1}^{n} \frac{\sum_{t \in obs(Z_i)}[Z_i(t) - X_i(t)]^2}{2\sigma_i^2} \qquad (1)$$

Where $n$ is the number of observations, $Z_i(t)$ is the i[th] observation at time $t$, $X_i(t)$ is the
corresponding simulation, $\sigma_i^2$ is the variance of the observations. The error is assumed to
independently follow a Gaussian distribution. This new set of parameter values will be accepted
if $J_{new}$ is smaller than $J_{old}$, the cost function with the previous set of accepted parameter values,
or the value, $\exp\left(-\frac{J_{new}}{J_{old}}\right)$, is larger than a random number selected from a uniform distribution
from 0 to 1 according to the Metropolis criterion (Liang et al., 2018; Luo et al., 2011; Shi et al.,



2018; Xu et al., 2006). Once the new set of parameter values is accepted, $J_{new}$ becomes $J_{old}$.
Those two phases of sampling will be iteratively executed until the number of sampling series set
in step 1 on preparation of DA is reached. Finally, the posterior distribution can be generated
from all the accepted parameter values.

MIDA realizes the execution of data assimilation according to the procedure described

above. First, MIDA uses a 'call' function to execute model simulations to get values of $X_i(t)$.
Observations $Z_i(t)$ and their variance $\sigma_i^2$ are already provided via the standardized data
exchange as described in step 1. Then, MIDA calculates $J_{new}$ according to equation 1 to decide
the acceptance of the current parameter values used in this simulation. If accepted, MIDA saves
this set of parameter values and associated $J_{new}$ values in $P_{accepted}$ and $J_{accepted}$ arrays
respectively and triggers new proposing phrase based on this set of accepted parameter values. If
not, MIDA discards this set of parameter values and generates another new set of parameter
values. MIDA saves the new parameter values generated in the proposing phrase to
"ParameterValue.txt", from which the model reads before execution of the next model
simulation. MIDA repeats the proposing and moving phases until the number of sampling series
is reached. At the end, MIDA selects the best parameter values through maximum likelihood
estimation and run model again using this set of values to get optimized simulation outputs
$X_i(t)$. Then MIDA saves the arrays of accepted parameters, associated errors, maximum
likelihood estimates (MLE), and optimized state variables $X_i(t)$ to four files,
"parameter_accepted.txt", "J_accepted.txt", "MLE.txt", and "OptimizedSimu.txt", respectively.

This execution of DA algorithm in MIDA enables users to conduct DA as a black box

and is independent of any particular model.


## 2.5 Step 3: Visualization


Step 3 is to visualize the results of DA in step 2. The end products of DA are accepted parameter
values, their associated $J_{new}$ values, the maximum likelihood estimates, and optimized
simulation results as saved in the output files. MIDA enables visualization of parameter posterior
probabilistic density distributions with a Python script. In the script, MIDA first read accepted
parameter values from "parameter_accepted.txt" file. Then, MIDA generates
posterior probabilistic density function (PPDF) for each parameter via 'kdeplot' function in the
'seaborn' package. The maximum likelihood estimates of parameters correspond to the peaks of
PPDF. The distinctive mode of PPDF indicates how well the parameter uncertainty is
constrained. Finally, MIDA visualizes the PPDF for all parameters in a figure using the
'matplotlib' package.

## 2.6 Implementation and architecture of MIDA


MIDA is equipped with a graphical user interface (GUI) and users can easily execute it through
an interactive window. Users can also run MIDA as a script program without the GUI.  MIDA is
written in Python (version 3.7). For the GUI-version, all relevant Python packages used in MIDA
are compiled together, thus users do not need to install them by themselves. For the non-GUI
version, users need to install Python 3.7 and relevant packages (i.e., numpy, shutil, subprocess,
matplotlib and seaborn). MIDA is compatible with model source codes written in multiple
programming language (e.g., Fortran, C/C++, C#, MATLAB, R, or Python). It is also
independent of multiple operation systems (e.g., Windows, Linux, MacOS). In addition, MIDA
is also able to run on high-performance computing (HPC) platforms via task management
systems (e.g., Slurm).



The architecture of MIDA is class-based and each class is designed to describe an object
(e.g., parameter, observations, etc.) with variables and operations. Five classes are defined in
MIDA: parameter, observation, initialization, MCMC algorithm and the main program. The
main program is the start of MIDA execution. It calls functions from all other classes to finish
three-step workflow. As described in section 2.2, parameter and observation classes contain
variables to be transferred in data exchanges via file I/O operations. These operations are
implemented using the 'numpy' package. The initialization class is to read 'namelist.txt' in step
1 on data preparation and to assign values for the variables in all other classes. Then the class of
MCMC algorithm conducts DA as described in step 2. In this step, the simulation operation uses
a 'call' function in 'subprocess' package to call model executable. At the start of model
simulation, MIDA writes new parameter values to the 'ParameterValue.txt' file in the 'working
path' directory specified in step 1 on data preparation. Then the model executable read parameter
values from the 'ParameterValue.txt' file and run. After model simulation, DA algorithm can
read the model outputs by the output filenames indicated in the output configuration file. After
DA, step 3 executes an additional Python script to read accepted parameter values and plot the
posterior distributions of parameters. The plotting operations uses 'matplotlib' and 'seaborn'
packages. The implementation of GUI uses pyQt5 toolkit to support interactive usage of MIDA.
Users can also run MIDA in a non-interactive way with a 'main.py' script to trigger the three-
step workflows.

**2.7 User information of MIDA**
In order to use MIDA, users need to prepare data and a model. The data to be used in MIDA are
prior ranges and default values of parameters, parameter covariances, output configuration file,





observations and their variances. They are organized in different files. Before running MIDA,
users need to specify their filenames as suggested in step 1. When users want to use different
data sets in DA, they can simply change filenames with the new data sets via GUI or in the
'namelist.txt' file. The model to be used in MIDA should have those to-be-estimated parameter
values not fixed in model source code rather than changeable through 'ParameterValue.txt' file.
MIDA writes new parameter values in each proposing phase during DA to the
'ParameterValue.txt' file, from which the model reads the parameter values to run the
simulation.

To calculate the cost function, $J$, we have to have a one-to-one match between

observations and model outputs. For example, phenology models in one of the application cases
of MIDA below generate discrete dates of leaf onset, which is a one-to-one match to the
observations of spring leaf onset. In this case, observation $Z_i(t)$ and model output $X_i(t)$ to be
used in calculation of $J$ is straightforward. In the application case for dynamic vegetation, the
data to be used are leaf area in six layers in a forest of 302 years old whereas the model simulates
leaf areas in eight layers from 0 to 800 years. To match observation, the model generates outputs
of leaf areas in six layers when simulated forest age reaches 302 years. This requires users to
prepare an output configuration file to instruct MIDA to read model outputs and re-organize their
outputs to match observation. The output configuration file starts with a single line listing an
observation filename and its corresponding output filenames. Following lines are an instruction
set to be operated on the output files signified above. Each instruction is to match one or
continuous elements in observation with elements in outputs with the same length. A blank line
means there are no further instructions. Then a new matching between another observation and
model outputs starts.



Once MIDA finishes the execution of data assimilation, users may need basic knowledge
to assess the performance of DA. For example, the acceptance rate, which is given by MIDA, is
the fraction of proposed parameter values that is accepted. Ideally, the acceptance rate should be
about 30 ~ 40% (Xu et al., 2006). A very low acceptance rate indicates that many new proposed
parameter values ($P_{new}$) are rejected because $P_{new}$ jumps too far away from the previously
accepted parameter values (Robert and Casella, 2013; Roberts et al., 1997). In this case, users are
suggested to reduce a jump scale in the proposing phase. On the other hand, a very high
acceptance rate is likely because $P_{new}$ moves slowly from the previously accepted parameter
values. Users may increase the jump scale.
In addition, DA usually requires a convergence test to examine whether posterior
distributions from different sampling series converge or not. Convergence test requires running
DA parallelly or in multiple times with different initial parameter values. MIDA provides a
Gelman-Rubin (G-R) test (Gelman and Rubin, 1992) for this purpose. To use the G-R test, users
need to prepare a file containing initial parameters values in different sampling series and
indicate its filename in the 'namelist.txt' file as described in step 1. If the G-R statistics
approaches one, the sampling series in DA is converged. When sampling series is converged, all
accepted parameter values are used to generate the posterior distributions.
There are three types of posterior distributions: bell-shape, edge-hitting, and flat. The
bell-shaped posterior distributions indicate that these parameters are well constrained. Their peak
values are the maximum likelihood estimates of parameter values. The flat posterior distributions
suggest that the parameters are not constrained due to the lack of relevant information in data.
The edge-hitting posterior distributions result from complex reasons. Users may change the prior





ranges to examine if those posterior distributions can be improved or examine correlations
among estimated parameters.

### 3.  Applications of MIDA

We applied MIDA to four groups of models, which are an ecosystem carbon cycle model, a
surrogate-based land surface model, nine phenology models, and a dynamic vegetation model,
respectively. These four cases demonstrate that MIDA is effective for stand-alone DA, flexible
to be applied to different models, and efficient for multiple model comparison.

**3.1 Case 1: Independent data assimilation with DALEC**

The first case study is to demonstrate that MIDA can be effective for independent data
assimilation with the data assimilation linked ecosystem carbon (DALEC) model (Williams et
al., 2005). DALEC has been used for data assimilation in several studies (Bloom et al., 2016; Lu
et al., 2017; Richardson et al., 2010; Safta et al., 2015; Williams et al., 2005). Previous studies all
incorporated data assimilation algorithms into DALEC, which requires invasive coding. This
case study is focused on reproducing the data assimilation results as in the study by Lu et al.
(2017) but with MIDA.

The version of DALEC used in this study is composed of six submodels (i.e.,

photosynthesis, phenology, autotrophic respiration, allocation, litterfall, and decomposition) to
simulate the carbon exchanges among five carbon pools (i.e., leaf, stem, root, soil organic matter
and litter) (Ricciuto et al., 2011). There are 21 parameters in DALEC, of which, 17 parameters
are derived from the six submodels and four parameters serve to initialize the carbon pools.
Table 2 summarizes the names, prior ranges and nominal values of these 21 parameters. The
observation is the Harvard Forest daily net ecosystem exchange (NEE) from year 1992 to 2006.



DALEC is coded in Fortran. In windows system, a gfortran compiler converts the model code to
an executable file (i.e., DALEC.exe).

Figure 2 is the GUI window of MIDA. We first set up a DA task as described in step 1

using the upper panel. In this application, the number of sampling series is set as 20,000. Once
users click the 'choose a directory' or 'choose a file' button, a new dialog window will pop up
and users are able to choose the directory or load files interactively. As describe in step 1 on
preparation of DA, the working path is where the outputs of DA and 'ParameterValue.txt' are
saved (e.g., C:/workingPath). After the output configuration file is loaded, the filenames of
model outputs, observations and their variance will be displayed in the window automatically.
This application only uses a 'NEE.txt' observation file. Similarly, after users load parameter
range file (e.g., a file named 'ParamRange.txt' contains three rows which are minimum,
maximum and default values of parameters), the content in this file is displayed as well. To
replace the current parameter range file loaded, users can simply upload another file. In this
application, the executive model file is 'DALEC.exe' with Fortran compiler in windows system.
Because we do not have parameter covariance information, this input is left blank. After 'save to
namelist file' is clicked, a 'namelist.txt' file containing all the inputs will be generated in the
working path.

After the DA task set up, we load the 'namelist.txt' file and click the 'run data

assimilation' button in the lower panel to trigger step 2 on execution of DA. A new dialog will
pop up to show the acceptance rate information and notify the termination of DA. Then we will
click the 'generate plots' button to visualize the posterior distributions of 21 parameters as
described in step 3.



Figure 3 shows that the simulation outputs using the optimized parameter values from
MIDA better fit with the observations than those using default parameter values. Figure 4 depicts
posterior distributions of the 21 parameters estimated from MIDA. More than half of the
parameters are constrained well with a unimodal shape. $X_{stem_{init}}$ and $X_{root_{init}}$ have a wide
occupation of the prior range, indicating that the observation data does not provide useful
information for them. The constrained posterior distributions in this study are similar to those
from the study in Lu et al. (2017). Note that MCMC estimates have a large variance and a low
convergence rate especially in high-dimensional problems, with a finite number of samples it is
not expected that two simulations would give exactly the same results.
**3.2 Case 2: Application of MIDA to a surrogate land surface model**
This case study is to examine the applicability of MIDA to a surrogate-based land surface model.
The original model is energy exascale earth system model the land component (ELM) (Ricciuto
et al., 2018). As ELM is computationally expensive (one forward model simulation takes more
than one day), a sparse-grid (SG) surrogate system was developed to reduce the computational
time (Lu et al., 2018). The forcing data for the surrogate model is half-hourly meteorological
measurements at Missouri Ozark flux site from 2006 to 2014. The observations that were used
for optimization are annual sums of net ecosystem exchange (NEE), annual averages of total leaf
area index and latent heat fluxes from 2006 to 2010. The eight parameters selected (Table 3) are
the most important parameters for the variations in outputs (Ricciuto et al., 2018). The model is
written in Python. A 'pyinstaller' library packages the model code into an executable file. The
iteration number in MIDA is 20,000.
Figure 5 shows posterior distributions of calibrated parameters. $c_{root}$, $SLA_{top}$,
$t_{leaffall}$, $GDD_{onset}$ are constrained well with a unimodal distribution. However, the distribution



of the rest 4 parameters (i.e., $N_{leaf}$, $CN_{root}$, $A_{r2l}$ and $Res_m$) cluster at near the edge. These
results match well with the study by Lu et al. (2018). As shown in Figure 6, the calibrated
parameters induce a performance improvement in simulating total leaf area index and NEE. For
latent heat, both the default and optimized simulation obtain good agreement with the
observation. These conclusions are also similar to those in Lu et al. (2018).

MIDA hides the detailed differences between models. For example, DALEC model in

case 1 is a process-based model to simulate ecosystem carbon cycle while surrogate-based ELM
in case 2 is an approximation of land surface model. They are also different in programming
language, simulation time, forcing data, etc. MIDA is able to deal with models with so many
different characteristics and hides these differences from users. Users only need to indicate the
filenames of the model to be used, its parameter range, the output configuration file, etc. in the
'namelist.txt' file. Thus, MIDA simplified the DA applications using different models.

**3.3 Case 3: Evaluation of multiple phenological models**
This study case uses nine phenological models (Yun et al., 2017) to demonstrate the applicability
of MIDA in model comparison. Five out of the nine models predict phenological events, such as
the day of leaf onset, using growing degree days, which are calculated as temperature
accumulation above a base temperature. The other four models consider two processes: chilling
effects of cold temperature on dormancy before budburst and forcing effects of warm
temperature on plant development. Each model uses different response functions to represent
chilling and forcing effects. The detailed model descriptions and associated parameter
information are in supplementary table.





Data are from the Spruce and Peatland Responses Under Climatic and Environmental
Change experiment (SPRUCE) (Hanson et al., 2017) located in northern Minnesota, USA. The
experiment consists of five-level whole-ecosystem warming (i.e., +0, +2.25, +4.5, +6.75, +9°C)
and two-level elevated $CO_2$ concentrations (i.e., +0, +500ppm). Dates of leaf onset were
observed with PhenoCam (Richardson et al., 2018) for tree species: *Picea mariana* and *Larix*
*laricina*. For the sake of demonstration of MIDA application, we only show DA results for *Larix*
*laricina* with +9°C warming treatment and +0 ppm $CO_2$ treatment from 2016 to 2018.
MIDA was used to compare performances of the nine models in reference to the same
observations of leaf onset dates after DA. We as users changed filenames of model executable
file (i.e., PhenoModels.exe), defined parameter ranges, and assigned the directory of working
path for each model. MIDA then estimated the optimized parameters and save the corresponding
best simulation outputs to the working path for each of the nine models. Figure 7 shows the best
simulation output of these nine models. The simulation output of the 6th, 7th, 8th, and 9th models
better fit the observation than the other models. It demonstrates that models that consider both
chilling and heating effects can achieve good simulations of the leaf onset dates.

**3.4 Case 4: Supporting data assimilation with a dynamic vegetation model**
This case study is to examine the efficiency of MIDA to integrate remote sensing data into a
dynamic vegetation model. The model used in this study is Biome Ecological strategy simulator
(BiomeE) (Weng et al., 2019). This model is to simulate vegetation demographic processes with
individual-based competition for light, soil water, and nutrients. Individual trees in BiomeE
model are represented by cohorts of trees with similar sizes. The light competition among
cohorts is based on their heights and crown areas according to the rule of perfect plasticity





approximation (PPA) model (Strigul et al., 2008). Each cohort has seven pools: leaves, roots,
sapwood, heartwood, seeds, nonstructural carbon and nitrogen. After carbon are assimilated into
plants via photosynthesis, the assimilated carbon enters to nonstructural carbon pool and is used
for plant growth (i.e., diameter, height, crown area) and reproduction according to empirical
allomeric equations (Weng et al., 2019). In this application, two parameters to be constrained
(Table 4) are annual productivity rate and annual mortality rate of trees.

Observations to be used in DA are leaf area indexes in six vertical heights (i.e., 0-5m, 6-

10m, 11-15m, 16-20m, 21-25m, and 26-30m) at Willow Creek study site, Wisconsin, USA. The
forest at the site is an upland deciduous broadleaf forest of around 302 years old. The
observations were from Global Ecosystem Dynamics Investigation (GEDI) acquired by a Light
Detection and Ranging (Lidar) laser system, which is deployed on the International Space
Station (ISS) by NASA in 2018 (Dubayah et al., 2020). The observations were first averaged
from three footprints and then leaf area indexes in the six canopy layers were standardized to be
summed up as one.

To use MIDA, we reorganized the simulation outputs to match observations as suggested

in section 2.6. The BiomeE model simulates leaf areas in eight layers (i.e., 0-5m, 6-10m, 11-
15m, 16-20m, 21-25m, 26-30m, 31-35m, and 36-40m) from 0 to 800 years. An output
configuration file was provided to post-process model outputs of leaf area indexes in six layers to
match observations at the forest age of 302 years. These simulated leaf area indexes in the six
canopy layers were also standardized to match standardized observations of leaf area indexes.
The observations and post-processed simulation outputs were saved to 'LAI.txt' and
'simu_LAI.txt' files, respectively. The two files are used in MIDA for data assimilation to
generate posterior distributions of estimated two parameters as showed in figure 8. The





optimized parameter values through maximum likelihood estimation are different from their
default values. Figure 9 compares the simulation outputs with optimized parameters estimated by
MIDA to those with default parameter values. After DA with GEDI data in MIDA, the
simulation accuracy of leaf area index is substantially improved especially in middle (16~20m)
and highest (26~30m) layers.

**4.  Discussion**
This study introduced MIDA as a model-independent tool to facilitate the applications of data
assimilation in ecology and biogeochemistry. The potential user community is ecologists with
limited knowledge of model programming and technical implementation of DA algorithms.
Several model-independent DA tools have already been developed, such as DART (Anderson et
al., 2009), openDA (Ridler et al., 2014),  PDAF (Nerger and Hiller, 2013) and PEST (Doherty,
2004), mainly for applications in research areas of hydrology, atmosphere, and remote sensing.
These DA tools either use gradient descent method, such as Levenburg-Marqurdt algorithm in
PEST, or Kalman Filter methods, such as EnKF in DART, openDA, and PDAF. The Levenburg-
Marqurdt algorithm is a local search method, which is hard to find global optimization solution
for highly nonlinear models. EnKF updates state variables and parameter values each time when
observations are sequentially assimilated, resulting discrete values of estimated parameters.
Jumps in estimated parameter values by EnKF make it very difficult to obey mass conservation
in biogeochemical models. In this study, we used the MCMC method in MIDA to generates
parameter values and their posterior distributions. MCMC is a widely used method in many DA
studies with biogeochemical models but has been applied to individual models with invasive
coding (Bloom et al., 2016; Hararuk et al., 2015; Liang et al., 2018; Luo and Schuur, 2020;





Ricciuto et al., 2011). MIDA is the first model-independent tool that uses the MCMC method for
DA.

Biogeochemical models are incorporating more detailed processes related to carbon and

nitrogen cycles (Lawrence et al. 2020). Complex biogeochemical models yield predictions with
great uncertainty (Frienlingstein et al. 2009 and 2014).  Data assimilation has been increasingly
used to estimate parameter values against observations and reduce uncertainty in model
prediction (Luo et al. 2016, Luo and Schuur 2020). However, current applications of DA are
almost all model dependent. It requires ecologists to write code to integrate DA algorithm with
models. The coding practice is a big technical challenge for ecologists with limited program
ability. The distinct advantage of MIDA is to enable ecologists to conduct model independent
DA. MIDA streamlines workflow of the three-step procedure for DA to enable users to conduct
DA without extensive coding. Users mainly need to provide numerical and character values for
data exchanges to transfer data (i.e., parameter values, simulation outputs, observations) between
the model and MIDA by a file named 'namelist.txt' or by interactive inputs via a GUI window
(Fig. 1).

We tested MIDA in four cases for its applicability to ecological models. The first case is

applied to DALEC model, which has been used in several data assimilation studies (Bloom et al.,
2016; Lu et al., 2017; Safta et al., 2015; Williams et al., 2005). The previous DA studies all used
invasive coding to incorporate DA algorithm into models. As demonstrated in this study, MIDA
was applied to DALEC without invasive coding but by providing the directory to save DA
results and filenames of DALEC model executable, parameter prior range, and output
configuration file through the 'namelist.txt' file or interactive inputs in the first preparation step
of the workflow. Then, MIDA run DA as a black box with DALEC before visualizing the DA





results. Next, we tested the applicability of MIDA a surrogate-based ELM model and a dynamic
vegetation model BiomeE. To switch the test case from DALEC to the surrogate-based ELM
model and the BiomeE model, we changed the filenames of model executable, parameter prior
range, and output configuration file in the 'namelist.txt' file for MIDA. This flexibility of MIDA
in switching models for DA makes it much easier for model comparisons. We tested this
capability of MIDA with nine phenological models to compare alternative model structures.
Similarly, MIDA enables efficient switches of observations to be assimilated into models. Users
only need to change filenames of observations in the output configuration file. This feature of
MIDA makes it easier to utilize abundant traits databases such as TRY (Kattge et al., 2020),
FRED (Iversen et al., 2017), etc. Moreover, this feature of MIDA also helps evaluating the
relative information content of different observations for constraining model parameters and
prediction (Weng and Luo, 2011). Consequently, MIDA can facilitate selection of the most
informative observations and then better guide data collections in filed experiments. Ultimately,
MIDA can aid ecological forecasting and help reduce uncertainty in model predictions (Huang et
al., 2018; Jiang et al., 2018).

Although MIDA helps users to get rid of model detail, users may still need basic

knowledge about the model outputs to prepare the output configuration file which is to match
model outputs to observations one-by-one (see Section 2.6). This effort of preparing the
correspondence between model outputs and observations for MIDA is not that difficult because
users are reading or writing a text file and most model developers will provide reference to help
understanding observations or model output files.





The current version of MIDA only incorporates Metropolis-Hasting sampling approach.
More MCMC methods (e.g., Hamiltonian Monte Carlo) may be incorporated into MIDA in the
future.

**5. Conclusions**

We developed MIDA to facilitate data assimilation for biogeochemical models. Traditional DA
studies require ecologists to program codes to integrate DA algorithms into model source codes.
The easy-to-use MIDA module enables ecologists to conduct model-independent DA without
extensive coding thus advancing the application of DA for ecological modeling and forecasting.
We demonstrated the capability of MIDA in four cases with a total of 12 ecological models.
These cases showed that MIDA is easy to perform for a variety of models and can efficiently
produce accurate parameter posterior distributions. Moreover, MIDA supports flexible usage of
different models and different observations in the DA analysis and allows a quick switch from
one model to another. This capability enables MIDA to serve as an efficient tool for model
intercomparison projects and enhancing ecological forecasting.

**Appendix A:** Nine phenological models
1.   Growing degree (GD)
The growing degree (GD) model is one of the most widespread phenological model to simulate
the date of leaf onset ($\widehat{D}$). In this study, the time scale is limited to daily based on observation
records. The kernel of GD is to calculate the growing degree days (GDD, $\sum_{d=D_s}^{\widehat{D}-1} \Delta d$) which is the
heat accumulation above a base temperature ($T_b$). For simplicity, the daily temperature ($T_d$) can
be approximated by the average of daily maximum and minimum temperatures. The heat





accumulation starts at day $D_s$, which is empirically estimated, and ends when GDD reaches a
forcing requirement threshold ($R_d$). Two parameters to be constrained are base temperature ($T_b$)
and the forcing requirement ($R_d$). Their default values and prior range are listed in Table A1.
$$\Delta d = \begin{cases} T_d - T_b \ if \ T_d > T_b \\ 0 \quad otherwise \end{cases} \quad (A1)$$
$$\sum_{d=D_s}^{\hat{D}-1} \Delta d < R_d \le \sum_{d=D_s}^{\hat{D}} \Delta d \quad (A2)$$
2.   Sigmoid function (SF)
Compared to the linear response function of GDD in GD model, the sigmoid function (SF)
model provides a non-linear function to better represent the non-linearity of the growth response
to heat accumulation. Three parameters to be constrained in DA are base temperature ($T_b$), the
forcing requirement ($R_d$) and temperature sensitivity ($S_t$). Their default values and prior range
are listed in Table A1.
$$\Delta d = \frac{1}{1+e^{S_t(T_d-T_b)}} \quad (A3)$$
$$\sum_{d=D_s}^{\hat{D}-1} \Delta d < R_d \le \sum_{d=D_s}^{\hat{D}} \Delta d \quad (A4)$$
3.   Beta function (BF)
In reality, the plant growth rate, as described with $\Delta d$, gradually increases up to a specific
temperature, then rapidly declines to a supra-optimal level. Such response can be well described
by a beta function with uni-modality and non-symmetrical shape. Three parameters are involved
in DA: minimum temperature ($T_n$), optimal temperature ($T_o$) and forcing requirement ($R_d$). The
other parameter values are fixed with empirical values. For example, maximum growth rate ($R_x$)
is set to one and maximum temperature ($T_x$) is assumed to be 45.
$$r_d = R_x \left(\frac{T_x-T_d}{T_x-T_o}\right)\left(\frac{T_d-T_n}{T_o-T_n}\right)^{\frac{T_o-T_n}{T_x-T_o}} \quad (A5)$$
$$\Delta d = \begin{cases} r_d \ if \ r_d > 0 \\ 0 \ otherwise \end{cases} \quad (A6)$$



$$\sum_{d=D_s}^{\widehat{D}-1} \Delta d < R_d \leq \sum_{d=D_s}^{\widehat{D}} \Delta d \quad (A7)$$
4.   Days transferred to standard temperature (DTS)
According to Arrhenius las, the relationship between growth rate and daily temperature $T_d$ can
be interpolated by the equation 8 (Ono and Konno, 1999). With a factor weighted by standard
temperature, the equation for DTS (Eq. A9) can better represent growth rate dependent on
temperatures. Three parameters considered in DA are: temperature sensitivity rate ($E_a$), standard
temperature ($T_s$) and forcing requirement ($R_d$).
$$k = e^{\frac{-E_a}{R \cdot T_d}} \quad (A8)$$
$$\Delta d = e^{\frac{E_a(T_d - T_s)}{R \cdot T_d \cdot T_s}} \quad (A9)$$
$$\sum_{d=D_s}^{\widehat{D}-1} \Delta d < R_d \leq \sum_{d=D_s}^{\widehat{D}} \Delta d \quad (A10)$$
5.   Thermal period fixed model (TP)
The difference between GD and TP models are heat accumulation occurs in a fixed time period
($D_n$). The day of leaf onset is the last day ($\widehat{D_s} + D_n$) when the accumulated heat reaches the
forcing requirement. The start day ($\widehat{D_s}$) of heat accumulation begins in day one and moves one
day forward each time to estimate Eq. (A12). Three parameters are involved in DA: the base
temperature ($T_b$), the period length ($D_n$) and the forcing requirement ($R_d$).
$$\Delta d = \begin{cases} T_d - T_b \ if \ T_d > T_b \\ 0 \quad otherwise \end{cases} \quad (A11)$$
$$R_d \leq \sum_{d=\widehat{D_s}}^{\widehat{D_s}+D_n} \Delta d \quad (A12)$$
6.   Chilling and forcing (CF)
Compared to GD, there is another distinctive chilling period for dormancy. CF model
sequentially calculates two accumulations in opposite directions: chilling accumulation and anti-
chilling accumulation. The start day of chilling accumulation ($D_s$) is implicitly set as 273.0



which is October $1^{st}$. The end day of chilling accumulation ($D_0$) is the beginning of anti-chilling
accumulation. Three parameters are considered in DA: the chilling requirement ($R_d^C$) and the
forcing requirement ($R_d^F$), the temperature threshold ($T_c$).
$$\Delta d = \begin{cases} T_d - T_c & if\ T_d \geq 0 \\ -T_c & otherwise \end{cases} \text{(A13)}$$

$$\Delta_d^C = \begin{cases} \Delta d & if\ \Delta d < 0 \\ 0 & otherwise \end{cases} \text{(A14)}$$

$$\Delta_d^F = \begin{cases} \Delta d & if\ \Delta d > 0 \\ 0 & otherwise \end{cases} \text{(A15)}$$

$$\sum_{d=D_s}^{D_0-1} \Delta_d^C > R_d^C \geq \sum_{d=D_s}^{D_0} \Delta_d^C \text{ (A16)}$$

$$\sum_{d=D_0}^{\widehat{D}-1} \Delta_d^F < R_d^F \leq \sum_{d=D_0}^{\widehat{D}} \Delta_d^F \text{ (A17)}$$

7.   Sequential model (SM)
The difference between CF and SM models is that SM used a beta function (Eq. A18) for the
calculation of chilling accumulation and adopted a sigmoid function (Eq. A20) for anti-chilling
accumulation. The detailed descriptions of these two functions can be referred to the
introductions of BF model and CF model. The maximum temperature is empirically set as
13.7695. Six parameters are constrained in DA: minimum temperature ($T_n$), optimal temperature
($T_o$), temperature sensitivity ($S_t$), forcing base temperature ($T_b$), chilling requirement ($R_d^C$), and
forcing requirement ($R_d^F$).
$$r_d = \left(\frac{T_x - T_d}{T_x - T_o}\right)\left(\frac{T_d - T_n}{T_o - T_n}\right)^{\frac{T_o - T_n}{T_x - T_o}} \text{ (A18)}$$

$$\Delta_d^C = \begin{cases} r_d & if\ r_d < 0 \\ 0 & otherwise \end{cases} \text{(A19)}$$

$$\Delta_d^F = \frac{1}{1 + e^{S_t(T_d - T_b)}} \text{ (A20)}$$

$$\sum_{d=D_s}^{D_0-1} \Delta_d^C > R_d^C \geq \sum_{d=D_s}^{D_0} \Delta_d^C \text{ (A21)}$$

$$\sum_{d=D_0}^{\widehat{D}-1} \Delta_d^F < R_d^F \leq \sum_{d=D_0}^{\widehat{D}} \Delta_d^F \text{ (A22)}$$



8.  Parallel model (PM)
Critical difference between PM and above two-step models is that the chilling and anti-chilling
accumulations happen simultaneously (Fu et al., 2012). In the earlier dates during chilling
period, only small fraction ($K_d$) of forcing (Eq. A25) will be accumulated. The maximum
temperature is empirically set as 15.3.  Seven parameters will be considered in DA: minimum
temperature ($T_n$), optimal temperature ($T_o$), temperature sensitivity ($S_t$), forcing base temperature
($T_b$), chilling requirement ($R_d^C$), forcing requirement ($R_d^F$), and a forcing weight coefficient ($K_m$).
$$r_d = \left(\frac{T_x - T_d}{T_x - T_o}\right)\left(\frac{T_d - T_n}{T_o - T_n}\right)^{\frac{T_o - T_n}{T_x - T_o}} \text{ (A23)}$$

$$\Delta_d^C = \begin{cases} r_d \ if \ r_d < 0 \\ 0 \ otherwise \end{cases} \text{ (A24)}$$

$$K_d = \begin{cases} K_m + (1 - K_m)\frac{\sum_{i=D_s}^{d} \Delta_i^C}{R_d^C} \ if \ \sum_{d=D_s}^{D_0 - 1} \Delta_d^C > R_d^C \\ 1 \qquad\qquad\qquad otherwise \end{cases} \text{ (A25)}$$

$$\Delta_d^F = \frac{K_d}{1 + e^{S_t(T_d - T_b)}} \text{ (A26)}$$

$$\sum_{d=D_s}^{D_0 - 1} \Delta_d^C > R_d^C \geq \sum_{d=D_s}^{D_0} \Delta_d^C \text{ (A27)}$$

$$\sum_{d=D_0}^{\widehat{D} - 1} \Delta_d^F < R_d^F \leq \sum_{d=D_0}^{\widehat{D}} \Delta_d^F \text{ (A28)}$$

9.  Alternating model (AM)
AM fixes the start date of chilling period ($D_s^C$) as November 1st and the start date of anti-chilling
period ($D_s^F$) as January 1st.  The difference between AM and the other models above is that the
forcing requirement is not a parameter value but is decided by the length of chilling days (Fu et
al., 2012). Five parameters to be constrained in DA are: chilling temperature ($T_c$), forcing base
temperature ($T_b$) and three coefficients ($a, b, c$) in calculation of forcing requirement.
$$\Delta_d^C = \begin{cases} 1 \ if \ T_d \leq T_c \\ 0 \ otherwise \end{cases} \text{ (A29)}$$

$$\Delta_d^F = \begin{cases} T_d - T_b \ if \ T_d > T_b \\ 0 \qquad otherwise \end{cases} \text{ (A30)}$$





$$R_d^C = \sum_{i=D_S^C}^{d} \Delta_i^C \quad (A31)$$

$$R_d^F = a + b \cdot e^{-c \cdot R_d^C} \quad (A32)$$

$$\sum_{d=D_S^F}^{\widehat{D}-1} \Delta_d^F < R_d^F \le \sum_{d=D_S^F}^{\widehat{D}} \Delta_d^F \quad (A33)$$


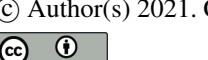



Table A1: A summary of parameters to be calibrated in nine phenological models. Their default
parameter value and prior parameter range are shown.

| Model | Parameter | Description | Unit | Default | Range |
|---|---|---|---|---|---|
| GD | $T_b$ | Base temperature | °C | 10 | [-5, 25] |
| | $R_d$ | Forcing requirement | °Cd | 35 | [0, 200] |
| SF | $T_b$ | Base temperature | °C | -1.5 | [-10, 25] |
| | $R_d$ | Forcing requirement | °C | 50 | [0, 500] |
| BF | $T_o$ | Optimal temperature | °C | 15 | [10, 35] |
| | $T_n$ | Minimum temperature | °C | 0 | [-10, 5] |
| | $R_d$ | Forcing requirement | °Cd | 11 | [0, 50] |
| DTS | $E_a$ | Temperature sensitivity rate | - | 250 | [1, 1500] |
| | $T_s$ | Standard temperature | °C | 10 | [-30, 40] |
| | $R_d$ | Forcing requirement | °Cd | 50 | [1, 200] |
| TP | $T_b$ | Base temperature | °C | 12.5 | [0, 30] |
| | $D_n$ | Period length | d | 25 | [0, 50] |
| | $R_d$ | Forcing requirement | °Cd | 20 | [0, 150] |
| CF | $R_d^C$ | Chilling requirement | °Cd | -124 | [-300, 0] |
| | $R_d^F$ | Forcing requirement | °Cd | 120 | [0, 300] |
| | $T_c$ | Chilling base temperature | °C | 5 | [0, 30] |
| SM | $T_n$ | Minimum temperature | °C | -20 | [-80, 0] |
| | $T_o$ | Optimal temperature | °C | 0 | [-26, 10] |
| | $S_t$ | Temperature sensitivity | - | -1.8 | [-5, 0] |
| | $T_b$ | Forcing base temperature | °C | 5 | [-5, 35] |
| | $R_d^C$ | Chilling requirement | °Cd | 20 | [0, 80] |
| | $R_d^F$ | Forcing requirement | °Cd | 20 | [0, 80] |
| PM | $T_n$ | Minimum temperature | °C | -20 | [-80, 0] |
| | $T_o$ | Optimal temperature | °C | 0 | [-26, 10] |
| | $S_t$ | Temperature sensitivity | - | -0.6 | [-1, 0] |
| | $T_b$ | Forcing base temperature | °C | 5 | [-5, 35] |
| | $R_d^C$ | Chilling requirement | °Cd | 11.35 | [0, 80] |
| | $R_d^F$ | Forcing requirement | °Cd | 44.01 | [0, 80] |
| | $K_m$ | Forcing weight coefficient | - | 0.2 | [0, 1] |
| AM | $T_c$ | Chilling base temperature | °C | 4.6 | [-10, 10] |
| | $T_b$ | Forcing base temperature | °C | 5 | [-5, 35] |
| | a | Coefficient for forcing adjustment | - | 11.51 | [0.01, 15] |
| | b | Coefficient for forcing adjustment | - | 88 | [0, 200] |
| | c | Coefficient for forcing adjustment | - | -0.01 | $[-1, -10^{-4}]$ |






*Code and data availability*. The code of MIDA is available at the GitHub repository
https://github.com/Celeste-Huang/MIDA (last access: Feb 2021). Data used in this study are
available at https://github.com/Celeste-Huang/MIDA/tree/main/Example.

*Video supplement*. A tutorial video of how to use MIDA is available at
https://github.com/Celeste-Huang/MIDA/tree/main/Videos

*Author contributions*. XH, IS, and YL designed the study. XH built the workflow of MIDA and
tested its capability in four cases. DL, DMR, and PJH provided data and model for the first and
second test cases. XL prepared models and ADR provided observations for the third case. EW
and SN helped to prepare data and model for the fourth case. XH, LJ, EH and YL analyzed the
results. All authors contributed to the preparation of the manuscript.

*Competing interests*. The authors declare that they have no conflict of interest.

*Acknowledgements*. This work was funded by subcontract 4000158404 from Oak Ridge National
Laboratory (ORNL) to the Northern Arizona University. ORNL is managed by UT-Battelle,
LLC, for the U.S. Department of Energy under contract DE-AC05-00OR22725.

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

920



Table1: Comparison among MIDA and available DA tools

| DA tool | Agnostic | DA algorithms | Global optima | Posterior distribution | Visualization |
|---|---|---|---|---|---|
| CCDAS | No | Automatic differentiation from Transformation of Algorithms in Fortran (TAF) | No | No | No |
| CARDAMOM | No | Markov Chain Monte Carlo | Yes | Yes | No |
| EcoPAD | No | Markov Chain Monte Carlo | Yes | Yes | Yes |
| OpenDA | No | EnKF, Ensemble Square-Root Filter, Particle Filter | Yes | Yes | No |
| DART | Yes | EnKF | Yes | Yes | No |
| PDAF | Yes | EnKF | Yes | Yes | No |
| PEST | Yes | Levenberg-Marquardt method | Rely on initial parameter values | No | No |
| MIDA | Yes | Markov Chain Monte Carlo | Yes | Yes | Yes |





Table 2: A summary of 21 parameters to be calibrated in DALEC model. The default parameter value and prior parameter range are shown.

| Parameter | Description | Unit | Default | Range |
|---|---|---|---|---|
| $GDD_{min}$ | Growing degree day threshold for leaf out | $^oC\ d$ | 100 | [10, 250] |
| $GDD_{max}$ | Growing degree day threshold for maximum LAI | $^oC\ d$ | 200 | [50, 500] |
| $LAI_{max}$ | Seasonal maximum leaf area index | - | 4 | [2, 7] |
| $T_{leaffall}$ | Temperature for leaf fall | $^oC$ | 5 | [0, 10] |
| $K_{leaf}$ | Rate of leaf fall | $d^{-1}$ | 0.1 | [0.03 0.95] |
| $NUE$ | N use efficiency | - | 7 | [1, 20] |
| $Res_{growth}$ | Growth respiration fraction | - | 0.2 | [0.05, 0.5] |
| $Res_m$ | Base rate for maintenance respiration | $\times 10^{-4}\ \mu mol\ m^{-2}d^{-1}$ | 1 | [0.1, 100] |
| $Q_{10mr}$ | Maintenance respiration T-sensitivity | - | 2 | [1, 4] |
| $A_{stem}$ | Allocation to plant stem pool | - | 0.7 | [0.1, 0.95] |
| $\tau_{root}$ | Root turnover time | $\times 10^{-4}\ d^{-1}$ | 5.48 | [1.1, 27.4] |
| $\tau_{stem}$ | Stem turnover time | $\times 10^{-5}\ d^{-1}$ | 5.48 | [1.1, 27.4] |
| $Q_{10hr}$ | Heterotrophic respiration T-sensitivity | - | 2 | [1, 4] |
| $\tau_{litter}$ | Base turnover for litter | $\times 10^{-3}\ umol\ m^{-2}d^{-1}$ | 1.37 | [0.548, 5.48] |
| $\tau_{som}$ | Base turnover for soil organic matter | $\times 10^{-4}\ umol\ m^{-2}d^{-1}$ | 9.13 | [0.274, 2.74] |
| $K_{decomp}$ | Decomposition rate | $\times 10^{-3}\ d^{-1}$ | 1 | [0.1, 10] |
| $LMA$ | Leaf mass per area | $gC\ m^{-2}$ | 80 | [20, 150] |
| $X_{stem_{init}}$ | Initial value for stem C pool | $\times 10^3\ gC$ | 5 | [1, 15] |
| $X_{root_{init}}$ | Initial value for root C pool | $gC$ | 500 | [100, 3000] |
| $X_{litter_{init}}$ | Initial value for litter C pool | $gC$ | 600 | [50, 1000] |
| $X_{som_{init}}$ | Initial value for soil organic C pool | $\times 10^3\ gC$ | 7 | [1, 25] |



Table 3: A summary of eight parameters to be calibrated in surrogate-based ELM model. The default parameter value and prior parameter range are shown.

| Parameter | Description | Unit | Default | Range |
|---|---|---|---|---|
| $c_{root}$ | Rooting depth distribution parameter | $m^{-1}$ | 2.0 | $[0.5, 4]$ |
| $SLA_{top}$ | Specific leaf area at canopy top | $m^2 gC^{-1}$ | 0.03 | $[0.01, 0.05]$ |
| $N_{leaf}$ | Fraction of leaf N in RuBisCO | - | 0.1007 | $[0.1, 0.4]$ |
| $CN_{root}$ | Fine root C:N ratio | - | 42 | $[25, 60]$ |
| $A_{r2l}$ | Allocation ratio of fine root to leaf | - | 1.0 | $[0.3, 1.5]$ |
| $Res_m$ | Base rate for maintenance respiration | $\times 10^{-6} \mu mol\ m^{-2} s^{-1}$ | 2.525 | $[1.5, 4]$ |
| $t_{leaffall}$ | Critical day length for senescence | $\times 10^4$ s | 3.93 | $[3.5, 4.5]$ |
| $GDD_{onset}$ | Accumulated growing degree days for leaf out | $^oC\ d$ | 800 | $[600, 1000]$ |

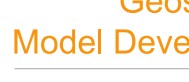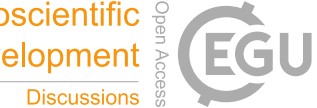



Table 4: A summary of two parameters to be calibrated BiomE model. The default parameter value and prior parameter range are shown.

| Parameter | Description | Unit | Default | Range |
|---|---|---|---|---|
| $V_{annual}$ | Annual productivity per unit leaf area | $kgC\ y^{-1}m^2$ | 0.4 | $[0.2, 2]$ |
| $M_{canopy}$ | Annual mortality rate in canopy layer | $y^{-1}$ | 0.02 | $[0.01, 0.08]$ |



**Figure captions**

**Figure 1**: The three-step workflow of Model Independent Data Assimilation (MIDA) module. The workflow includes data preparation, execution of data assimilation (DA), and visualization. The data preparation step is to provide all the formatted essential data for DA via user input. The execution step is to calibrate parameter values towards a constrained posterior distribution with the fusion of observations. The visualization step is to diagnose the effects of DA. Rhombus in orange represents user-input data. Rectangle represents procedures and document/multidocument shape is for data files in computers. Dashed lines indicate locations of data. Solids lines indicate data flow pathways. With the three-step workflow, DA is agnostic to specific models and users will be released from technical burdens.

**Figure 2**: the GUI-MIDA window includes two panels. The upper panel is to set up a data assimilation task. Inputs can be loaded and applied to the step 1 on data preparation for DA. The lower panel is to run DA as described in step 2 and visualize the posterior distributions of parameters in step 3.

**Figure 3**: Comparison between the simulated daily net ecosystem exchange (NEE) by DALEC and the observed NEE at Harvard Forest from 1992 to 2006. Red circles represent modeled NEE with the optimized parameter values and green circles represent simulated NEE with the original parameter values. Simulations of DALEC are substantially improved after data assimilation in comparison with those before data assimilation.

**Figure 4**: Comparison between posterior distributions (red line) and default values (gray dash line) of the 21 parameters in DALEC. The peak in posterior distribution is the constrained parameter value from maximum likelihood estimation. This distinctive mode and its divergence from the default value indicates the effects of DA. Most parameters are well constrained, and some are far different from the original values.

**Figure 5**: Comparison between posterior distributions (red line) and default values (gray dash line) of the eight parameters in surrogate-based ELM. The peak in posterior distribution is the constrained parameter value from maximum likelihood estimation. This distinctive mode and its divergence from the default value indicates the effects of DA. Most parameters are well constrained, and some are far different from the original values.

**Figure 6**: Comparison between the simulated NEE, total leaf area index, latent heat flux by surrogate-based ELM and the observed ones at Missouri Ozark flux site from 2006 to 2014. The





blue lines indicate the observations, and their 95% confidence interval is in the dashed area. The green and red lines indicate the simulations with default parameter values and optimized values respectively. Simulations are generally improved after DA for all these three variables.

**Figure 7**: Comparison between the simulated growth date by 9 phenology models after DA and the observed growth date for *Larix laricina* with +9°C treatment at SPRUCE site from 2016 to 2018. Colored number indicates different models and shape represents different year. Overall, model 6,7,8,9 achieve better performance after DA.

**Figure 8**: Comparison between posterior distributions (red line) and default values (gray dash line) of the two parameters in BiomeE. The peak in posterior distribution is the constrained parameter value from maximum likelihood estimation. This distinctive mode and its divergence from the default value indicates the effects of DA. All parameters are well constrained and different from their original values.

**Figure 9**: Comparison between the simulated leaf area index (LAI) by BiomeE and the observed NEE at Willow Creek. Circles represent modeled NEE with the optimized parameter values and triangles represent simulated NEE with the original parameter values. Simulations of LAI are substantially improved after data assimilation in comparison with those before data assimilation.



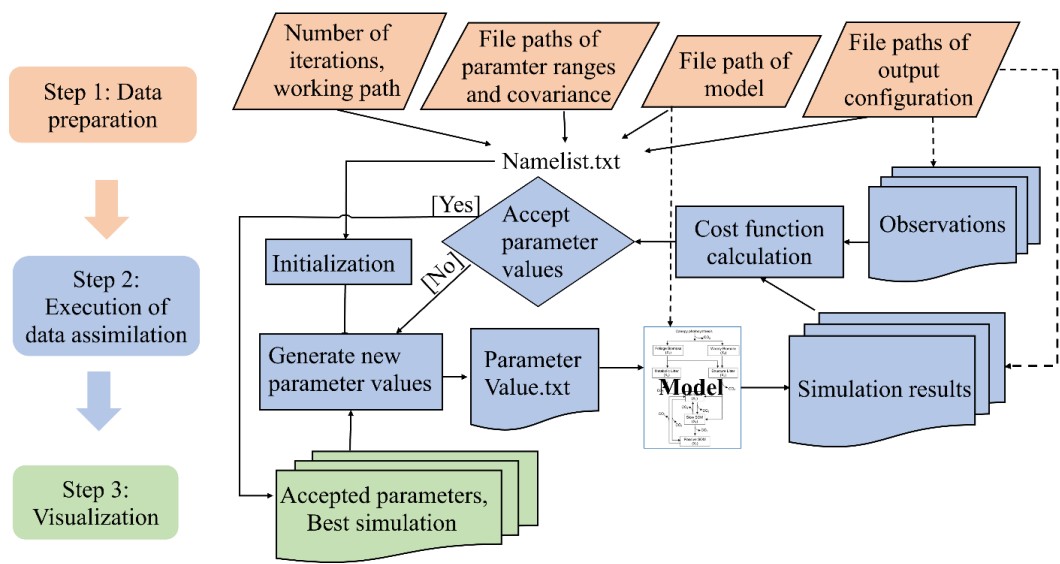

**Figure 1**: The three-step workflow of Model Independent Data Assimilation (MIDA) module. The workflow includes data preparation, execution of data assimilation (DA), and visualization. The data preparation step is to provide all the formatted essential data for DA via user input. The execution step is to calibrate parameter values towards a constrained posterior distribution with the fusion of observations. The visualization step is to diagnose the effects of DA. Rhombus in orange represents user-input data. Rectangle represents procedures and document/multidocument shape is for data files in computers. Dashed lines indicate locations of data. Solids lines indicate data flow pathways. With the three-step workflow, DA is agnostic to specific models and users will be released from technical burdens.



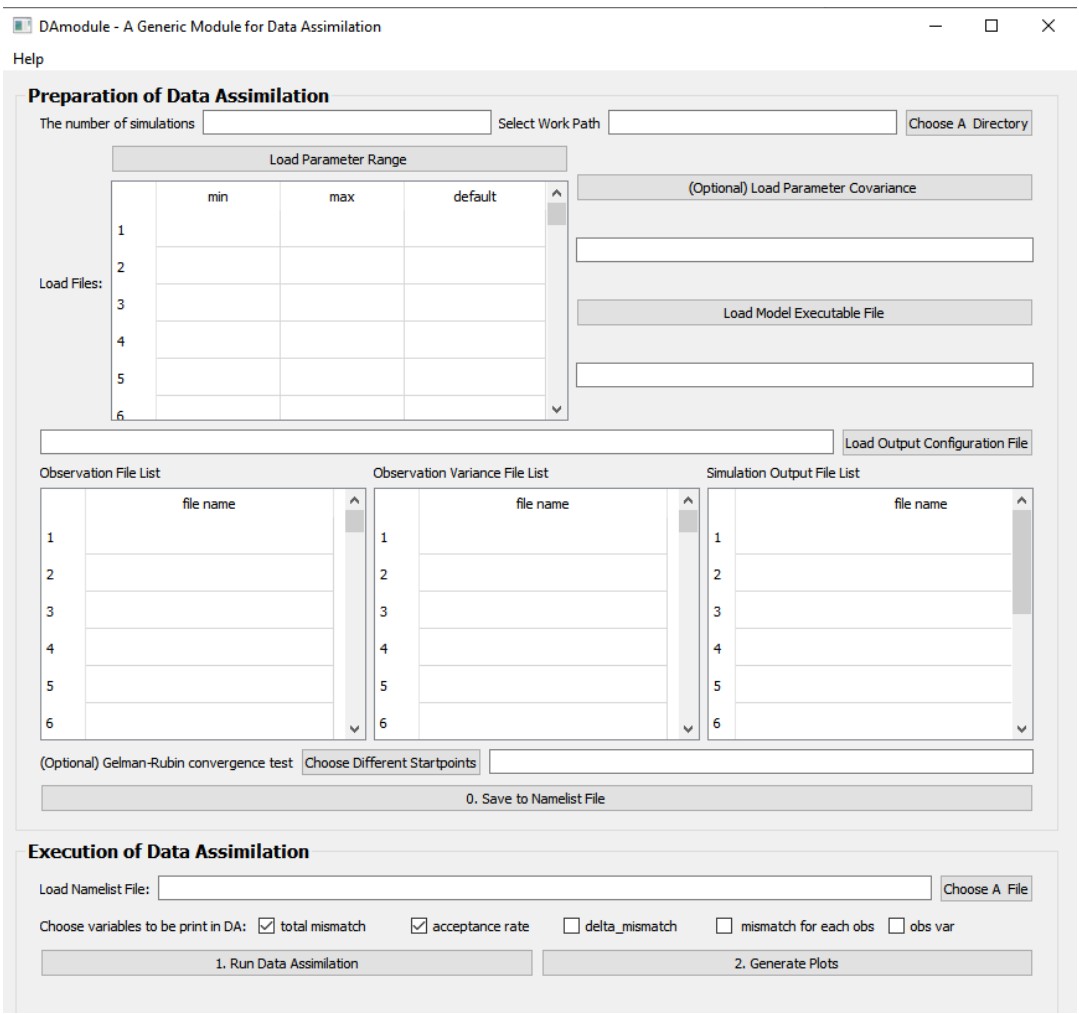

**Figure 2**: the GUI-MIDA window includes two panels. The upper panel is to set up a data assimilation task. Inputs can be loaded and applied to the step 1 on data preparation for DA. The lower panel is to run DA as described in step 2 and visualize the posterior distributions of parameters in step 3.



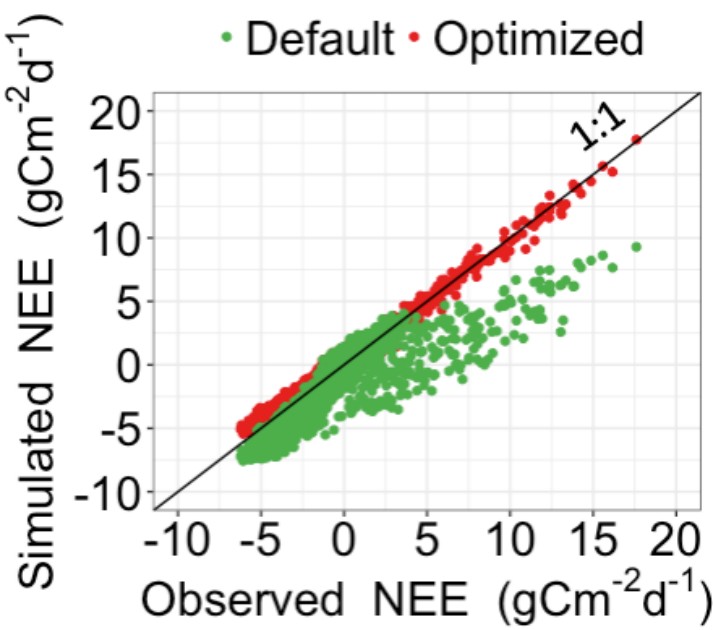

**Figure 3**: Comparison between the simulated daily net ecosystem exchange (NEE) by DALEC and the observed NEE at Harvard Forest from 1992 to 2006. Red circles represent modeled NEE with the optimized parameter values and green circles represent simulated NEE with the original parameter values. Simulations of DALEC are substantially improved after data assimilation in comparison with those before data assimilation.





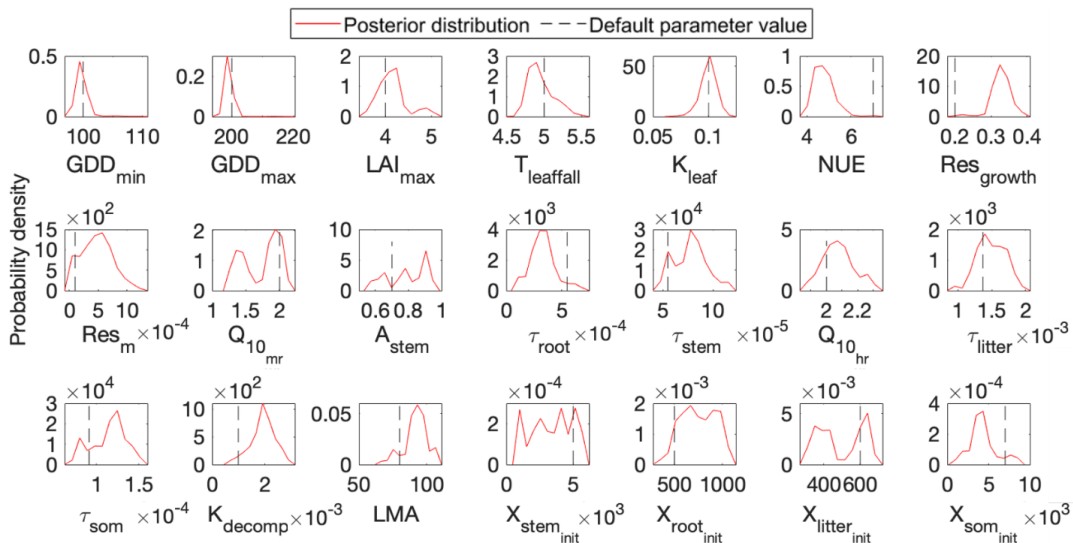

**Figure 4**: Comparison between posterior distributions (red line) and default values (gray dash line) of the 21 parameters in DALEC. The peak in posterior distribution is the constrained parameter value from maximum likelihood estimation. This distinctive mode and its divergence from the default value indicates the effects of DA. Most parameters are well constrained, and some are far different from the original values.





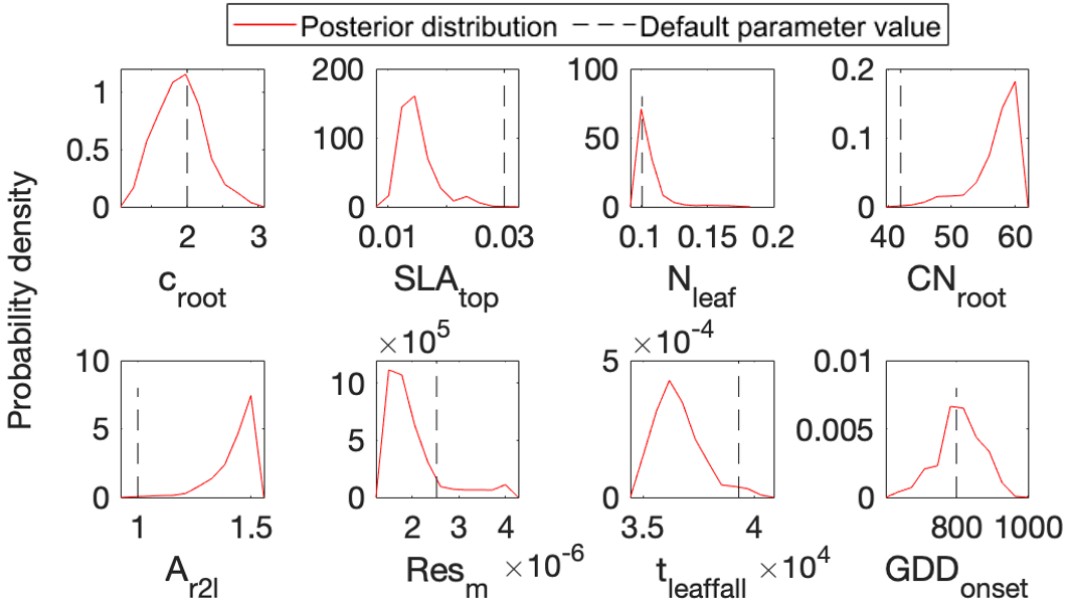

**Figure 5**: Comparison between posterior distributions (red line) and default values (gray dash line) of the eight parameters in surrogate-based ELM. The peak in posterior distribution is the constrained parameter value from maximum likelihood estimation. This distinctive mode and its divergence from the default value indicates the effects of DA. Most parameters are well constrained, and some are far different from the original values.



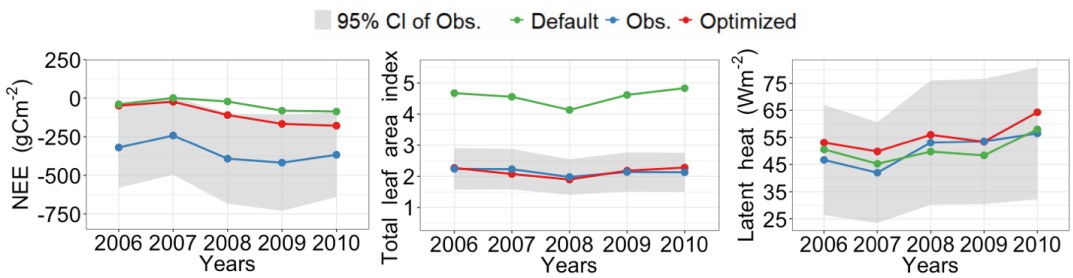

**Figure 6**: Comparison between the simulated NEE, total leaf area index, latent heat flux by surrogate-based ELM and the observed ones at Missouri Ozark flux site from 2006 to 2014. The blue lines indicate the observations, and their 95% confidence interval is in the dashed area. The green and red lines indicate the simulations with default parameter values and optimized values respectively. Simulations are generally improved after DA for all these three variables.





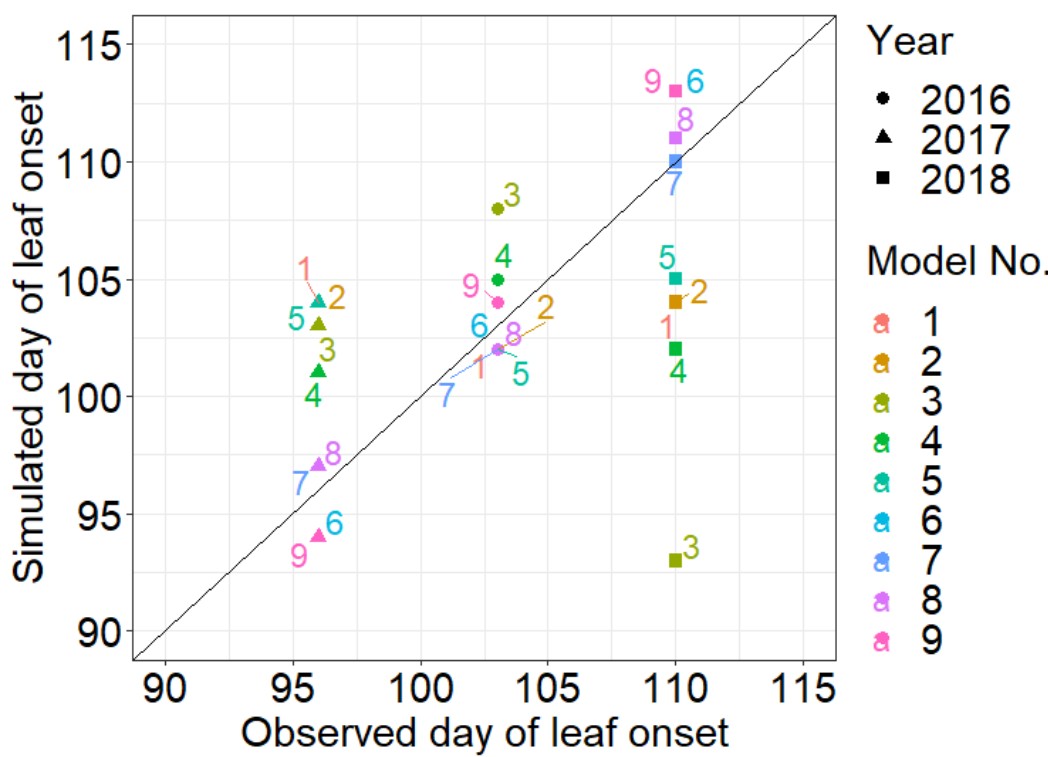

**Figure 7**: Comparison between the simulated growth date by 9 phenology models after DA and the observed growth date for *Larix laricina* with +9°C treatment at SPRUCE site from 2016 to 2018. Colored number indicates different models and shape represents different year. Overall, model 6,7,8,9 achieve better performance after DA.



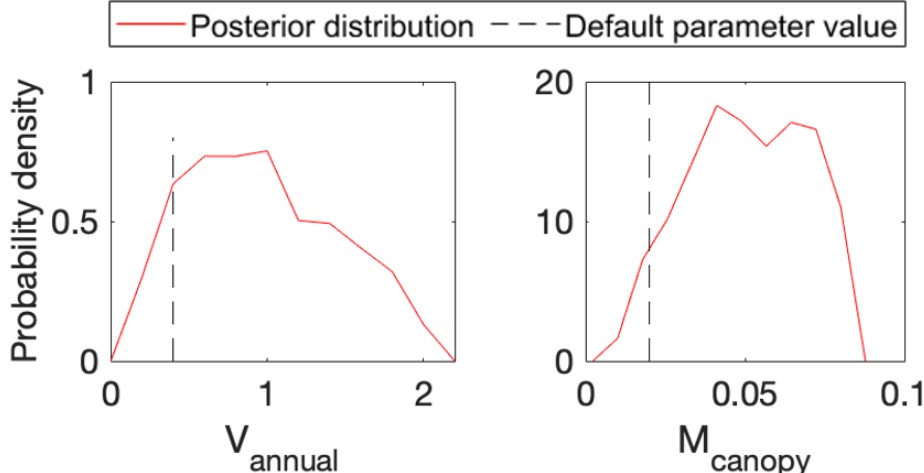

**Figure 8**: Comparison between posterior distributions (red line) and default values (gray dash line) of the two parameters in BiomeE. The peak in posterior distribution is the constrained parameter value from maximum likelihood estimation. This distinctive mode and its divergence from the default value indicates the effects of DA. All parameters are well constrained and different from their original values.





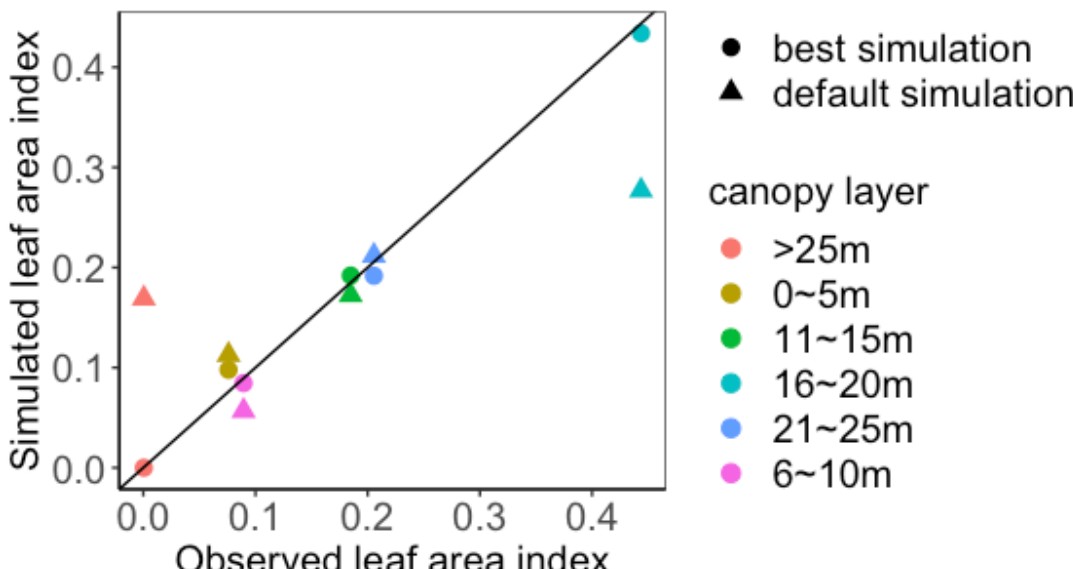

**Figure 9**: Comparison between the simulated leaf area index (LAI) by BiomeE and the observed NEE at Willow Creek. Circles represent modeled NEE with the optimized parameter values and triangles represent simulated NEE with the original parameter values. Simulations of LAI are substantially improved after data assimilation in comparison with those before data assimilation.