# Peer review of "A Model-Independent Data Assimilation (MIDA) module and its applications in ecology"

_Geoscientific Model Development, 2021_

## Author Comment (AC3)

**Manuscript**

**Responses to Reviewers**

Dear editor and reviewers,

Thanks very much for taking your time to review this manuscript. We really appreciate all your comments and suggestions, which are very helpful for us to improve the manuscript. We have made a thorough revision to address the comments and questions. Please find our point-to-point responses below.

**Reviewers' Comments to the Authors:**

**Reviewer 2**

1. *The statistical model used for model calibration in Section 2.4 "Step: Execution of data assimilation" is not well defined. It's quite unclear to me what the authors are using for model calibration therefore it is difficult to evaluate the calibration exercises themselves. I'm not convinced that you are really showing posterior distributions in the figures because of the description of the algorithm in the methods. What is the actual statistical model used for model calibration?*

Thanks for the comments and the question. The statistical method for model calibration in Section 2.4 is Metropolis-Hasting algorithm. It is a sampling-based algorithm including a proposing step and a moving step. The proposing step (L231-237) is to generate new set of parameter values and the moving step (L238-248) is to decide whether to accept these new parameter values or not. The posterior distribution is generated from all accepted parameters (L250). The significant peak in the posterior distribution indicates the parameters are well constrained (L360-366).

2. *My second major comment is that Fer 2018 and PEcAn were entirely left out of this manuscript but should certainly be included in several places.*

We appreciate this constructive comment. Including Fer 2018 and PEcAn is a great addition and We have added Fer et al. (2018) and PEcAn in L101.

*3. Line 26: what kinds of states?*

We are sorry for the unclear description. We changed "state" to "states of ecosystems" in L26.

*4. Line 38: I think there's a word missing "model ... the land component"*

Thanks for your question. The Energy, Exascale, Earth System Model (E3SM) is an Earth system modeling project sponsored by the US Department of Energy (E3SM Project, 2018). E3SM land model (ELM) is the land and energy component of the earth system model. To be clearer, we added a colon between 'model' and 'the land component' in L39.

*5. Line 42: Doesn't the easy implementation potentially make this more 'black box'?*

Thanks for pointing it out. We removed "black-box" to avoid misleading in L42.

*6. Line 58: Citation for invasive coding?*

Thanks for the great suggestion. Invasive coding, a concept from Java programming language, means to modify current code to incorporate new algorithms or features. In this study, invasive coding means programming the DA algorithm into the model source code. We added a citation for invasive coding in L58.

*7. Line 75: Missing link sentence between ", 2009). ... DA was"*

Yes, the link is missing. We added "In the study by Liang et al. (2018)" to better link the two sentences in L75.

*8. Line 78: Not sure about "data-worth"*

We apologize for the vagueness. To avoid confusion, we removed the redundant phrase "for data-worth analysis" in L78.

*9. Line 100: Include Predictive Ecosystem Analyzer (PEcAn)*

This is a great suggestion. We added PEcAn and the citation of Fer et al., (2018) in L101.

10. *Line 120-121: Not sure what is meant by this sentence*

Sorry for the unclear statement. This sentence is to express a point that changes in estimated parameter values by EnKF each time when a data point is assimilated usually do not reflect a reality of biogeochemical cycles in the real world. That is, parameter values of biogeochemical cycles in the real world do not suddenly change at a time point when data is assimilated by EnKF. We have modified the sentence to clarify this point.

*11. Line 131: Nice!*

Thanks.

*12. Line 176: Hinders?*

Thanks for pointing it out. We have corrected the typo to "hides" in L191.

*13. Line 178: How does MIDA know how to write out model specific configurations?*

We hope we have understood your question correctly. Generally, MIDA does not write out configurations for a specific model. Instead, MIDA uses a 'call' function written in Python to run the model executable first and does not require model-specific forcing or other configurations (L253). Then, the data exchanges in the communication between MIDA and the model are realized by file I/O operations and MIDA does not write out model-specific configurations, either (L196-203). Taking parameter values as an example, MIDA reads the parameter range from a file "namelist.txt" that is provided by users. According to the parameter range, MIDA gets the number, maximum limit and minimum limit of the parameters. Based on this information, MIDA generates new parameter values and writes them to a file for the model executable to read. Third, all model-specific information is provided by users. For example, users need to indicate the file names of parameter range, observations, and model outputs in the "namelist.txt" or via GUI. Users also need to prepare a model-specific output configuration file to instruct how to map model outputs with each observation. Section 2.7 describes such information in details. Appendix B and Appendix C provides an example of the output configuration file and an example of the namelist.txt file, respectively.

*14. Line 183 transfers -> transfer*

We have corrected the typo to "transfer" in L198 as suggested.

*15. Line 184: organize -> organizes*

The typo has been corrected to "organizes" in L199 as suggested.

*16. Line 184: So the "different files" are like lookup tables?*

Thanks for the great question. The 'different files' in the original L184 are to save parameter ranges, parameter values, simulation outputs, observations and their covariances, and an output configuration to match between observations and simulation outputs. Different from PEcAn, there are no lookup tables in MIDA. PEcAn uses a lookup table to find the right data (e.g., PFT) for a specific model. MIDA defines a prototype class for data (i.e., parameter, observation or simulation output). These classes are initialized to adapt a specific model according to the data files loaded when MIDA is running. These coding methods are object-orient programming and dynamic initialization, which are commonly used in Java programming language. All this information is available in L204-215.

*17. Line 194: Would be cool to have an illustration of dynamic initialization \**

Thanks for the suggestion. Object-orient programming and dynamic initialization in L207 are concepts from Java programming. As readers of this manuscript are most likely ecologists who may not have advanced knowledge about programming, it would make it easier if we avoid such technical details.

*18. Line 209: In think you mean "inference"*

We have corrected the typo to "inference" in L224 as suggested.

*19. Line 210: Before talking about the sampling algorithm, it would be good for the reader to know about the model formulation like the likelihood, prior, etc.*

We thank the reviewer for the valuable suggestion. In Section 2.1, we introduce general concepts about data assimilation methods. We revised to include description of likelihood, prior, and posterior distribution in L141-154.

*20. Line 245: What kind of MLE? What's the model formulation? What are you maximizing over here? Why do you get maximum likelihoods and posteriors?*

Thanks for the great questions. All these questions are related to data assimilation (DA), which is to estimate parameter posterior distributions and constrain parameter values through assimilating observations into the model. Based on Bayes' theorem, estimating parameter posterior distribution is transferred to maximize the likelihood function (Eq. 1,2), which is negatively proportional to the mismatch between simulation outputs and observations (Eq. 3). Meanwhile, maximum likelihood estimator of Eq. 2 can also help estimate the optimal parameter values. The distinctive mode of the posterior distributions indicates whether the parameter uncertainty is well constrained. All these information has been added to L141-154 in Section 2.1 to clarify the relationship between likelihood, posterior, and MLE in DA.

*21. Line 250: At this point, I'm still confused how you define your drivers and model settings in general? For a particular model you usually have a parameter file that gets read by the executable. So are you rewriting those parameter files in step 2?*

We apologize for the confusions. For the first question, model simulation is independent of MIDA and MIDA uses a 'call' function from the subprocess package in Python to execute model simulation. Taking DALEC model in the first case study as an example, MIDA calls executable file of DALEC, which has already defined the directory where forcings (i.e., temperature, vapor pressure deficit, carbon dioxide concentration, etc.) are read to execute. The data exchange between MIDA and the model is model-specific (i.e., parameter values, simulation outputs, observations and their variances) and users only need to provide the paths and names of these data files in the 'namelist.txt' file or via GUI. Appendix C is an example of a 'namelist.txt' file. Section 2.3 in the manuscript has introduced how to realize these model-specific data exchanges in MIDA. Section 2.7 has introduced how users prepare the 'namelist.txt' file.

For the second question, MIDA reads parameter range from a file indicated in the 'namelist.txt' file, which is provides by users. Based on the parameter range, MIDA repeatedly generates new set of parameter values during the DA and writes these parameter values to the 'ParameterValue.txt' in the work path that users chose in the 'namelist.txt'. Model executable needs to read this file to get new parameter values. Users can also change the name of 'ParameterValue.txt' in the 'namelist.txt' according to the need of a specific model. L259-261

and L320-324 described how MIDA writes parameter values to a file, from which the model reads these parameter values to run simulation.

*22. Line 263: Make sure to cite all software packages.*

Thanks for the kind reminder. We added citations of all Python packages used in MIDA in L287.

*23. Line 314: Helpful examples. Really curious where 302 years of leaf area data come from!*

We first generated simulation results of the BiomeE model with forest ages ranging from 0 to 800 years, then compared the simulated forest height or vertical structure to GEDI-derived observations and determined the forest age is 302 yrs. According to the empirical knowledge, the forest is around 300~350 years old and a variation of 50 years is acceptable as because forests are almost equilibrated.

*24. Line 343: "complex reasons" is somewhat vague*

We are sorry about the vague description. We added specific descriptions "such as improper prior parameter range" for the reasons leading to edge-hitting posterior distribution in L363. In addition, we included other possible reasons to explain the results.

*25. Line 499: Citation?*

Thanks for the suggestion. We added a citation of Gao et al. (2011) in L520.

*26. Line 504: Not sure about "first" unless you make your definition of model agnostic more specific*

We are sorry about the unclear statement. We changed the sentence to "Compared to the model-independent DA tools mentioned above, MIDA is the first that uses the MCMC method for DA.'" in L524. It echoes with L514-515.

*27. Figure 1: Very nice!*

Thanks.

*28. Figure 6: the Cis appear homogenous. Can MIRA deal with heteroskedasticity?*

Thanks for your question. MIDA uses MCMC algorithms which requires homogenous variance. MIDA is also flexible enough to incorporate other algorithms that can deal with data of heteroskedasticity.

*29. Figure 7: the colors in the legend have an extra character that should be removed*

As suggested, we removed the extra character in the legend of model no. in Fig. 7.

*30. Figure 4, 5, and 8: these posterior distributions do not look converged to me. Could you also include the mcmc chains as well here or in the supplements?*

Thanks for your question. These cases study are to repeat published DA results to demonstrate the capability of MIDA. Taking Fig. 4 as an example, the constrained posterior distributions are similar to those from the original study in Lu et al. (2017). In addition, MIDA has already incorporated algorithms to support Gelman-Robin convergence test of multiple MCMC chains as described in L335.

---

## Author Response (AR1)

**Manuscript**

**Responses to Reviewers**

Dear editor and reviewers,

Thanks very much for taking your time to review this manuscript. We really appreciate all your
comments and suggestions, which are very helpful for us to improve the manuscript. We have
made a thorough revision to address the comments and questions. Please find our point-to-point
responses below.

Note that line numbers in our responses are all referred to these in the revised manuscript
MIDA_GMD_revised-clean.pdf (the clean version).

**Reviewers' Comments to the Authors:**

**Reviewer 1**

*1. I would like a little more detail of is contained within the "black box" placed in the SI to give*
*those who are interested this information.*

Thanks for the suggestion. The black box in the manuscript refers to the execution of data
assimilation, which is described in sections 2.1 and 2.4. We hope the reviewer will find the
description is enough for readers to understand data assimilation. Moreover, sections 2.1 and 2.4
cite papers for readers to learn more about the method.

*2. I wonder about the trade-offs between the usually very fast exchange of information achieved*
*when writing an interface vs the more user friendly approach described here? Not an*
*objection to your approach but genuinely curious.*

This is a great question. We appreciate it. Generally, MIDA requires longer computation time
than the embedded data assimilation (DA) algorithms. The time difference depends on how to
call model simulation. Taking DALEC model in the first study as an example, the time cost for
the embedded algorithm is 24 mins while MIDA takes 52 mins to finish DA. Thus, we recommend the embedded algorithm for complex models with high computational demand while

MIDA is more suitable for beginners of DA users with models that are less complex. We have added this information about computation efficiency of MIDA in the discussion section of the manuscript (L569-572). The added sentence is "Generally, MIDA requires longer time to run

DA than the embedded DA algorithm, because MIDA calls model simulation as an external executable rather than a function embedded. Thus, we recommend MIDA for beginners of DA

users with models that are less complex."

Corresponding code is added to the Code Availability and is available in Zenodo repository (https://doi.org/10.5281/zenodo.4891319).

*3. What isn't quite so clear is the details of how MIDA knows which information in the existing*

*model output files corresponds to its observations. For example, the namelist.txt must*

*contain information on the variable names used to describe the observations and their*

*corresponding output variable generated by the model? These must still vary depending on*

*the model being used? A screen shot showing the interface which is populated with an*

*example would make this really clear.*

Thanks the reviewer for the great questions and suggestion. The information in the model output files that corresponds to its observations is in an output configuration file (e.g., config.txt), which notifies MIDA how to map model outputs to the observations. Users need to prepare the configuration file because, as the reviewer has mentioned, the configurations (or mapping functions) vary depending on the model being used. The output configuration file is described in

L335-342. As suggested, we added a screenshot of the output configuration file in Appendix B

(as shown below) and also added a link of Appendix B in L342: "An example of output configure file is available in Appendix B.".

**"Appendix B:** An example of output configuration file

Output configuration file (e.g., config.txt) is to indicate the directories of observations and simulation output files as well as how they map to each other. Figure B1 is an example of the output configuration file. There are three blocks of functions to map simulation outputs to observed GPP, RE, and NEE. The blocks of mapping functions are separated by a blank line.

Each mapping block starts with the directories of one observation, its observation variance and model outputs, which are separated by a hash key. If there is no observation variance available, users can ignore this directory. If multiple simulation outputs are used to correspond to one observation, the directories of simulation outputs are separated by a comma. The rest of the mapping block describes how to map simulation outputs to observations. The simu_map variable is simulation output after mapping. The simuList variable saves the simulation outputs specified in the first line. Taking the third mapping block in Fig. B1 as an example, simuList[0] saves contents in simuNEE_1.txt and simuList[0][0:365] saves the first 365 elements in this file.

[Figure]

Figure B1: An example of output configuration file"

*4.  The models need to be able to read the parameters from a file. The MIDA framework must*

*then be able to write out the proposed parameters in a unique format for each model, is that*

*correct?*

Yes, that is correct. We have described how MIDA writes new parameter values to a file

'ParameterValue.txt', from which the model reads to execute simulations in L261-263 ("MIDA

saves the new parameter values generated in the proposing phrase to "ParameterValue.txt", from which the model reads before execution of the next model simulation.") and L322-326 ("The model to be used in MIDA should have those to-be-estimated parameter values not fixed in model source code rather than changeable through 'ParameterValue.txt' file. MIDA writes new parameter values in each proposing phase during DA to the 'ParameterValue.txt' file, from which the model reads the parameter values to run the simulation. ").

*5. L322-330: Could you add a link to further details in SI for this section? The reason I ask is*

*that Haario et al., (2001) steps based on the weighted (e.g. beta) combination of the*

*multivariate Gaussian and a minimum step size scaled by a value drawn from a Gaussian*

*distribution of mean = 0, sd = 1. The multivariate Gaussian being derived from the*

*covariance matrix for the parameters adjusted by an optimal scaling parameter (e.g. 2.38 /*

*npars^0.5). The weighting between the two steps (beta ~0.05) and the minimum step size. So*

*which of these variables (or something else entirely) for example is you "jump scaling"?*

We thank the reviewer for this suggestion. Haario et al. (2001) introduced an adapted Metropolis algorithm, in which the proposal distribution is tuned along the search according to the covariance calculated from previous samples. The Metropolis-Hasting (MH) algorithm in this study uses a fixed Gaussian proposal distribution, in which the covariance is provided from test runs. A parameter covariance is not provided, the MH algorithm uses a uniform proposal distribution instead following this equation: $C_{new} = C_{old} + r \times (C_{max} - C_{min})/D$, where $r$ is a random number uniformly distributed in $[-0.5, 0.5]$, $C_{max}$ and $C_{min}$ are the maximum and minimum limits of parameters, respectively, $D$ is a scalar controlling the proposing step size.

Users can change the value of $D$ in the 'namelist.txt' file.

The above content has been described in L232-239:"The proposing phase generates a new set of parameter values based on the starting point for the first iteration or current accepted parameter values in the following iterations. If parameter covariance ($cov_{param}$) is specified in step 1 on data preparation, this proposing phase will draw new parameter values ($C_{new}$) within the prior ranges from a Gaussian distribution $N(C_{old}, cov_{param})$ where $C_{old}$ is the predecessor set of parameter values. Without parameter covariance, new set of parameter values will be generated from a uniform distribution within the prior ranges (Xu et al., 2006). "

To avoid the misleading by the citation of Haario et al. (2001), we corrected the citation to

Metropolis et al. (1953) and Hastings (1970) in L230. We also added a citation of Xu et al.

(2006) in L238 as Appendix B of Xu et al. (2006) explained the MH algorithm in detail.

The paragraph reviewer is asking about (L343-351) mainly described how to adjust the acceptance rate, which is a critical index to assess the performance of DA. And more details can be found in Xu et al. (2006), of which we have cited. So, we believe these would be adequate for readers to understand the method.

*6. I like the inclusion of a screenshot of the software but I think it would be useful to have an*

*example which has been filled in to help guide the potential user. Alternatively showing an*

*example of the namelist.txt might be informative.*

Thanks for the suggestion. We added a screenshot of the namelist.txt in Appendix C for the first case study with DALEC model (as shown below). A link to the Appendix C is also provided in

L321 ("Figure C1 is an example of the 'namelist.txt' file for a data assimilation study with the

DALEC model.").

"**Appendix C:** An example of the namelist.txt file

The Fig. C1 shows an example of the namelist.txt for the first study case with the DALEC

model. Users need to prepare the namelist.txt before execution of data assimilation (DA) either manually or via GUI. Below describes the content in the namelist.txt. Detailed explanation or tutorials are available in the Zenodo repositories at the end of the appendixes.

'workpath' is the directory where the MIDA executable are saved. 'nsimu' is the number of iterations in execution of data assimilation. 'J_default' is the default mismatch (i.e., cost function) to be compared in the first moving phase of data assimilation. 'ProposingStepSize'

controls the jump scale in the proposing phase of data assimilation. Users can increase or decrease this value to adjust the acceptance rate to be in a range from 0.2 to 0.5. 'paramFile' is the directory of a csv file saving parameter-related information such as parameter range.

'obsList' saves the directories of observations. Multiple observations are separated by semicolon.

Similarly, 'obsVarList' saves the directories of observation variance in the same order as that of

'obsList'. 'simuList' saves the directories of simulation outputs corresponding to the observations. With GUI, MIDA reads directories in the output configuration file (e.g., config.txt)

which users provide and assign values for 'obsList','obsVarList', and 'simuList' in the namelist.txt automatically. In this case, if the directories of observations change, users only need to modify the output configuration file and generate the namelist.txt again with GUI-based

MIDA.

'paramValue' is the directory of a txt file where MIDA writes out new set of parameter values for model execution in each iteration of data assimilation. Its default value is

'ParameterValue.txt' under the workpath specified in the first line of the namelist.txt. 'model'

saves the directory of model executable. 'nChains_convergeTest' indicates whether to conduct

German-Rubin (G-R) convergence test or not. If G-R test is used, its values is the number of multiple MCMC chains. If not, its value is zero. 'convergeTest_startsFile' is the directory of a csv file saving default parameter values as the start points in multiple MCMC chains.

'outConvergenceTest' saves the results of G-R test. If 'nChains_ConvergeTest' is zero, both values of 'convergeTest_startsFile' and 'outConvergenceTest' are empty. 'DAresultsPath' is the directory saving the results of DA whose directories are also listed in the following six lines:

'outJ' for the accepted mismatches; 'outC' for the accepted parameter values; 'outRecordNum'

for the number of accepted parameter values; 'outBestSimu' for the best simulation outputs with the optimal parameter values; 'outBestC' for the optimal parameter values. For MIDA without

GUI, 'display_plot' indicates whether or not to visualize the posterior distributions after DA.

[Figure]

**Figure C1**. An example of the 'namelist.txt' file. In order to use MIDA, users need to prepare data and a model and specify their file names and directories in the 'namelist.txt' file. "

7.  *This doesn't really impact the validity of the paper but just something I noticed and wanted*

*to raise as it should really be clarified. The DALEC model is stated as having 5 C pools but*

*also to having a Growing Degree Days phenology model. However, the Williams et al.,*

*(2005) model doesn't have phenology model (i.e. continuous allocation / evergreen). DALEC*

*was split into deciduous and evergreen versions in Fox et al., (2009) as part of the reflex*

*project adding a 6th pool and the GDD model. The example DALEC code provided on the*

*MIDA Github shows a alternate version of the model where leaf C is not dependent on GPP*

*(and thus the system is not mass balanced). This is a distraction from the main point of*

*demonstrating your DA system. Please make the origin of the code clear as it doesn't match*

*that found in the citations given.*

We apologize for the inconsistence. The version of DALEC model we used in this study is the version described by Lu et al. (2017).  It origins from Williams et al. (2005) but with some structural modifications. For example, the version of DALCE model by Lu et al. (2017)

incorporates the phenology submodel developed by Ricciuto et al. (2011). Compared to the version of DALEC used in Fox et al. (2009), the model used in this study works for deciduous species and the plant labile pool is removed for simplification. We corrected the citation to Lu et al. (2017) in L375.

In the code for DALEC in this manuscript, GPP is first consumed in autotrophic respiration, i.e., growth respiration (RG) and maintenance respiration (EM), and then is allocated to three vegetation pools, i.e., foliage (VEG_POOLS(1)), wood (VEG_POOLS(2)), and root (VEG_POOLS(3)). The variable NPP2 in L138 in the code is NPP minus change in leaf mass (CF_DELTA) which is used to update foliage pool. NPP2 in L209-210 in the code is used to update wood and root pools. Therefore, the sum of the changes in the three vegetation pools equals to NPP. Therefore, the DALEC model in this study has C mass balance. More detailed information is in Lu et al. (2017) which we cited in L375.

```
! get autotrophic respiration
call Ra(VEG_POOLS, CF_DELTA, GPP, RG, RM)
NPP  = GPP-RG-RM
NPP2 = NPP
if (CF_DELTA .gt. 0) NPP2 = NPP-CF_DELTA
```

```
! allocate carbon to vegetation pools
if (decid) VEG_POOLS(1) = VEG_POOLS(1)+CF_DELTA ! leaf
VEG_POOLS(2) = VEG_POOLS(2)+astem*NPP2           ! stem
VEG_POOLS(3) = VEG_POOLS(3)+(1.0d0-astem)*NPP2    ! root
```

8. *L140: "DA is a statistical approach..." – there are many different algorithms for DA whether for state update or parameter estimation (in this case). I think it would be clearer refer to it as an "approach". I can see that you are trying to talk about your specific approach so maybe "The DA approach embeded within MIDA..."*

We apologize for the confusion. Currently, only Metropolis Hasting algorithm is embedded in MIDA, but MIDA is open to incorporate many other DA algorithms. Therefore, it would be more appropriate to use "approach" rather than "The DA approach embedded within MIDA". We have changed "DA is a statistical algorithm" to "DA is a statistical approach" as suggested in L145.

9. *L176: "hinders" or "hides"?*

It should be "hides". We have corrected this typo in L193. The sentence is now "In MIDA, the process of data exchange calls a model executable file which hides the details of model code."

10. *L454: "This model simulates..."*

We have changed as suggested in L475. The revised sentence is now "This model simulates vegetation demographic processes with individual-based competition for light, soil water, and nutrients."

**Reviewer 2**

1. *The statistical model used for model calibration in Section 2.4 "Step: Execution of data assimilation" is not well defined. It's quite unclear to me what the authors are using for model calibration therefore it is difficult to evaluate the calibration exercises themselves. I'm not convinced that you are really showing posterior distributions in the figures because of*

*the description of the algorithm in the methods. What is the actual statistical model used for*

*model calibration?*

Thanks for the comments and the question. The statistical method for model calibration in

Section 2.4 is Metropolis-Hasting algorithm. It is a sampling-based algorithm including a proposing step and a moving step. The proposing step (L232-239) is to generate new set of parameter values and the moving step (L240-250) is to decide whether to accept these new parameter values or not. The posterior distribution is generated from all accepted parameters (L252-253). The significant peak in the posterior distribution indicates the parameters are well constrained (L280-281, L360-366).

*2. My second major comment is that Fer 2018 and PEcAn were entirely left out of this*

*manuscript but should certainly be included in several places.*

We appreciate this constructive comment. Including Fer 2018 and PEcAn is a great addition and

We have added  PEcAn in L104 where we describe all DA workflow systems. The sentence is

"A number of tools have been developed to facilitate DA applications (Table 1) but many of them are model dependent, such as the Carbon Cycle Data Assimilation Systems (CCDAS)

(Rayner et al., 2005; Scholze et al., 2007), the Carbon Data Model Framework (CARDAMOM)

(Bloom et al., 2016), the Ecological Platform for Assimilating Data (EcoPAD) into model (Huang et al. 2019) and Predictive Ecosystem Analyzer (PEcAn) (LeBauer et al., 2013).".

We also cited Fer 2018 in L82-84 as an example of the advanced algorithms to boost applications of DA: "In spite of powerful applications of DA to ecological research, computational cost is a major hurdle, especially with complex models. Fer et al. (2018)

developed a Bayesian model emulation to reduce the time cost of DA from 112h to 6h with the simplified Photosynthesis and Evapotranspiration model. ".

*3. Line 26: what kinds of states?*

We are sorry for the unclear description. We changed "state" to "states of ecosystems" in L26.

The sentence is now "An accurate prediction of future states of ecosystems depends on not only model structures but also parameterizations.".

*4. Line 38: I think there's a word missing "model … the land component"*

Thanks for your question. The Energy, Exascale, Earth System Model (E3SM) is an Earth system modeling project sponsored by the US Department of Energy (E3SM Project,

2018). E3SM land model (ELM) is the land and energy component of the earth system model.

To be clearer, we added a colon between 'model' and 'the land component' ("a surrogate-based energy exascale earth system model: the land component (ELM)") in L38 and L422.

*5. Line 42: Doesn't the easy implementation potentially make this more 'black box'?*

Thanks for pointing it out. We removed "black-box" to avoid misleading in L42. The updated sentence is "Additionally, the easy implementation and model-independent feature of MIDA

breaks the technical barrier of applications of data-model fusion in ecology."

*6. Line 58: Citation for invasive coding?*

Thanks for the great suggestion. Invasive coding, a concept from Java programming language, means to modify current code to incorporate new algorithms or features. In this study, invasive coding means programming the DA algorithm into the model source code. We added a citation ("Walls et al., 2005") for invasive coding in L59.

*7. Line 75: Missing link sentence between ", 2009). … DA was"*

Yes, the link is missing. We added "In the study by Liang et al. (2018)" to better link the two sentences in L75. The sentences are modified as "Second, DA can be used to select alternative model structures to better represent ecological processes (Liang et al., 2018; Van Oijen et al.,

2011; Shi et al., 2018; Smith et al., 2013; Williams et al., 2009). In the study by Liang et al.

(2018), DA was used to evaluate four models. And a two-pool interactive model was selected after DA to best represent SOC decomposition with priming."

*8. Line 78: Not sure about "data-worth"*

We apologize for the vagueness. To avoid confusion, we removed the redundant phrase "for data-worth analysis" in L78. The sentence is now "Additionally, DA can be applied to locate the most informative data to reduce uncertainty, thus guiding the sensor network design. (Keenan et al., 2013; Raupach et al., 2005; Shi et al., 2018; Williams et al., 2005)."

*9. Line 100: Include Predictive Ecosystem Analyzer (PEcAn)*

This is a great suggestion. We added PEcAn in L104 where all DA workflow systems are introduced. The sentence is modified as "A number of tools have been developed to facilitate DA

applications (Table 1) but many of them are model dependent, such as the Carbon Cycle Data

Assimilation Systems (CCDAS) (Rayner et al., 2005; Scholze et al., 2007), the Carbon Data

Model Framework (CARDAMOM) (Bloom et al., 2016), the Ecological Platform for

Assimilating Data (EcoPAD) into model (Huang et al. 2019) and Predictive Ecosystem Analyzer (PEcAn) (LeBauer et al., 2013)."

10. *Line 120-121: Not sure what is meant by this sentence*

Sorry for the unclear statement. This sentence is to express a point that changes in estimated parameter values by EnKF each time when a data point is assimilated usually do not reflect a reality of biogeochemical cycles in the real world. That is, parameter values of biogeochemical cycles in the real world do not suddenly change at a time point when data is assimilated by

EnKF. We have modified the sentence in L124-126 to clarify this point. The sentence is revised as "The sudden changes in estimated parameter values at time points when data are assimilated by EnKF usually do not reflect reality of biogeochemical cycles in the real world."

*11. Line 131: Nice!*

Thanks.

*12. Line 176: Hinders?*

Thanks for pointing it out. We have corrected the typo to "hides" in L193. The sentence is now

"In MIDA, the process of data exchange calls a model executable file which hides the details of model code.".

*13. Line 178: How does MIDA know how to write out model specific configurations?*

We hope we have understood your question correctly. Generally, MIDA does not write out configurations for a specific model. Instead, MIDA uses a 'call' function written in Python to run the model executable first and does not require model-specific forcing or other configurations (L255). Then, the data exchanges in the communication between MIDA and the model are realized by file I/O operations and MIDA does not write out model-specific configurations, either (L198-205). Taking parameter values as an example, MIDA reads the parameter range from a file "namelist.txt" that is provided by users. According to the parameter range, MIDA

gets the number, maximum limit and minimum limit of the parameters. Based on this information, MIDA generates new parameter values and writes them to a file for the model executable to read. Third, all model-specific information is provided by users. For example, users need to indicate the file names of parameter range, observations, and model outputs in the

"namelist.txt" or via GUI. Users also need to prepare a model-specific output configuration file to instruct how to map model outputs with each observation. Section 2.7 describes such information in details. Appendix B and Appendix C, which are listed below, provides an example of the output configuration file and an example of the namelist.txt file, respectively.

"**Appendix B:** An example of output configuration file

[revised manuscript text omitted]

*14. Line 183 transfers -> transfer*

We have corrected the typo to "transfer" in L200 as suggested. The sentence is revised as "This is because data exchange between DA algorithm and model uses memory to transfer items such as parameter values.".

*15. Line 184: organize -> organizes*

The typo has been corrected to "organizes" in L201 as suggested. The sentence is modified as "Instead, MIDA organizes items in data exchange using different files.".

*16. Line 184: So the "different files" are like lookup tables?*

Thanks for the great question. The 'different files' in the L201 are to save parameter ranges, parameter values, simulation outputs, observations and their covariances, and an output configuration to match between observations and simulation outputs. Different from PEcAn, there are no lookup tables in MIDA. PEcAn uses a lookup table to find the right data (e.g., PFT) for a specific model. MIDA defines a prototype class for data (i.e., parameter, observation or simulation output). These classes are initialized to adapt a specific model according to the data files loaded when MIDA is running. These coding methods are object-orient programming and dynamic initialization, which are commonly used in Java programming language. All this information is available in L209-216.

*17. Line 194: Would be cool to have an illustration of dynamic initialization ***

Thanks for the suggestion. Object-orient programming and dynamic initialization in L209 are concepts from Java programming. As readers of this manuscript are most likely ecologists who may not have advanced knowledge about programming, it would make it easier if we avoid such technical details. In the manuscript, we cite two papers on these concepts in case that readers would like to know more about them in L209. The sentence is "MIDA uses the concepts of object-orient programming (Mitchell and Apt, 2003) and dynamic initialization (Cline et al., 1998) in computer science to provide a homogenous way to create various observation types from a unified prototype class."

*18. Line 209: In think you mean "inference"*

We have corrected the typo to "inference" in L226 as suggested. The revised sentence is "Data assimilation usually uses some types of sampling algorithms, such as Markov chain Monte Carlo (MCMC), to generate posterior parameter distribution under a Bayesian inference framework (Box and Tiao, 1992).".

*19. Line 210: Before talking about the sampling algorithm, it would be good for the reader to know about the model formulation like the likelihood, prior, etc.*

We thank the reviewer for the valuable suggestion. In Section 2.1, we introduce general concepts about data assimilation methods. We revised to include description of likelihood, prior, and posterior distribution in L145-166. The revised paragraphs are shown below.

**"2.1 Bayes's theorem and DA**

Based on Bayes' theorem, DA is a statistical approach to constrain parameter values and estimate their posterior density distributions through assimilating observations into a model. The posterior density distributions $p(C|Z)$ of parameters $C$ for a given observation $Z$ can be obtained from *prior* density distributions $p(C)$ and the likelihood function $p(Z|C)$:

$$p(C|Z) \propto p(Z|C)p(C) \tag{1}$$

The *prior* density distribution $p(C)$ is assumed as a uniform distribution over the parameter range. And the likelihood function is negatively proportional to a cost function, $J$ as:

$$p(Z|C) \propto exp(-J) \tag{2}$$

The cost function measures the misfit between simulation outputs and observations and is described in more detail in section 2.4. The posterior density distributions $p(C|Z)$ is estimated from sampling parameter values to maximize the likelihood function $p(Z|C)$ or minimize the cost function $J$. DA usually uses a sampling technique, such as Markov chain Monte Carlo (MCMC) in this MIDA. The MCMC algorithm successively generates a new set of parameter values from the prior parameter ranges and requires model run with these new parameter values. Then the cost function is calculated to determine whether this new set of parameter values will be accepted or not according to the Metropolis-Hastings criterion (see more description in section 2.4). All accepted parameter values are used to generate posterior distributions where the distinctive mode indicates the parameter uncertainty is well constrained. Meanwhile, we derive maximum likelihood estimates (MLEs) of parameters from the posterior density distributions.

MIDA realizes model-independent Bayesian-based DA to estimate posterior density distributions and MLEs of parameters via data exchanges between a given model and DA algorithm. "

*20. Line 245: What kind of MLE? What's the model formulation? What are you maximizing over here? Why do you get maximum likelihoods and posteriors?*

Thanks for the great questions. All these questions are related to data assimilation (DA), which is to estimate parameter posterior distributions and constrain parameter values through assimilating observations into the model. Based on Bayes' theorem, estimating parameter posterior distribution is transferred to maximize the likelihood function (Eq. 1, 2), which is negatively proportional to the mismatch between simulation outputs and observations (Eq. 3). Meanwhile, maximum likelihood estimator of Eq. 2 can also help estimate the optimal parameter values. The distinctive mode of the posterior distributions indicates whether the parameter uncertainty is well constrained. All these information has been added to L145-163 in Section 2.1 to clarify the relationship between likelihood, posterior, and MLE in DA as shown below.

"**2.1 Bayes's theorem and DA**

Based on Bayes' theorem, DA is a statistical approach to constrain parameter values and estimate their posterior density distributions through assimilating observations into a model. The posterior density distributions $p(C|Z)$ of parameters $C$ for a given observation $Z$ can be obtained from *prior* density distributions $p(C)$ and the likelihood function $p(Z|C)$:

$$p(C|Z) \propto p(Z|C)p(C) \tag{1}$$

The *prior* density distribution $p(C)$ is assumed as a uniform distribution over the parameter range. And the likelihood function is negatively proportional to a cost function, $J$ as:

$$p(Z|C) \propto exp(-J) \tag{2}$$

The cost function measures the misfit between simulation outputs and observations and is described in more detail in section 2.4. The posterior density distributions $p(C|Z)$ is estimated from sampling parameter values to maximize the likelihood function $p(Z|C)$ or minimize the cost function $J$. DA usually uses a sampling technique, such as Markov chain Monte Carlo (MCMC) in this MIDA. The MCMC algorithm successively generates a new set of parameter values from the prior parameter ranges and requires model run with these new parameter values. Then the cost function is calculated to determine whether this new set of parameter values will be accepted or not according to the Metropolis-Hastings criterion (see more description in section 2.4). All accepted parameter values are used to generate posterior distributions where the distinctive mode indicates the parameter uncertainty is well constrained. Meanwhile, we derive maximum likelihood estimates (MLEs) of parameters from the posterior density distributions.

MIDA realizes model-independent Bayesian-based DA to estimate posterior density distributions and MLEs of parameters via data exchanges between a given model and DA algorithm."

*21. Line 250: At this point, I'm still confused how you define your drivers and model settings in general? For a particular model you usually have a parameter file that gets read by the executable. So are you rewriting those parameter files in step 2?*

We apologize for the confusions. For the first question, model simulation is independent of MIDA and MIDA uses a 'call' function from the subprocess package in Python to execute model simulation (L255). Taking DALEC model in the first case study as an example, MIDA calls executable file of DALEC, which has already defined the directory where forcings (i.e., temperature, vapor pressure deficit, carbon dioxide concentration, etc.) are read to execute. The data exchange between MIDA and the model is model-specific (i.e., parameter values, simulation outputs, observations and their variances) and users only need to provide the paths and names of these data files in the 'namelist.txt' file or via GUI. We added Appendix C, which is listed at the end, to show an example of a 'namelist.txt' file. Section 2.3 in the manuscript has introduced how to realize these model-specific data exchanges in MIDA. Section 2.7 has introduced how users prepare the 'namelist.txt' file.

For the second question, MIDA reads parameter range from a file indicated in the 'namelist.txt' file, which is provides by users. Based on the parameter range, MIDA repeatedly generates new set of parameter values during the DA and writes these parameter values to the 'ParameterValue.txt' in the work path that users chose in the 'namelist.txt'. Model executable needs to read this file to get new parameter values. Users can also change the name of 'ParameterValue.txt' in the 'namelist.txt' according to the need of a specific model. L262-263 and L322-326 described how MIDA writes parameter values to a file, from which the model reads these parameter values to run simulation.

"**Appendix C:** An example of the namelist.txt file

The Fig. C1 shows an example of the namelist.txt for the first study case with the DALEC model. Users need to prepare the namelist.txt before execution of data assimilation (DA) either manually or via GUI. Below describes the content in the namelist.txt. Detailed explanation or tutorials are available in the Zenodo repositories at the end of the appendixes.

'workpath' is the directory where the MIDA executable are saved. 'nsimu' is the number of iterations in execution of data assimilation. 'J_default' is the default mismatch (i.e., cost function) to be compared in the first moving phase of data assimilation. 'ProposingStepSize' controls the jump scale in the proposing phase of data assimilation. Users can increase or decrease this value to adjust the acceptance rate to be in a range from 0.2 to 0.5. 'paramFile' is the directory of a csv file saving parameter-related information such as parameter range. 'obsList' saves the directories of observations. Multiple observations are separated by semicolon.

Similarly, 'obsVarList' saves the directories of observation variance in the same order as that of 'obsList'. 'simuList' saves the directories of simulation outputs corresponding to the observations. With GUI, MIDA reads directories in the output configuration file (e.g., config.txt) which users provide and assign values for 'obsList','obsVarList', and 'simuList' in the namelist.txt automatically. In this case, if the directories of observations change, users only need to modify the output configuration file and generate the namelist.txt again with GUI-based MIDA.

'paramValue' is the directory of a txt file where MIDA writes out new set of parameter values for model execution in each iteration of data assimilation. Its default value is 'ParameterValue.txt' under the workpath specified in the first line of the namelist.txt. 'model' saves the directory of model executable. 'nChains_convergeTest' indicates whether to conduct German-Rubin (G-R) convergence test or not. If G-R test is used, its values is the number of multiple MCMC chains. If not, its value is zero. 'convergeTest_startsFile' is the directory of a csv file saving default parameter values as the start points in multiple MCMC chains. 'outConvergenceTest' saves the results of G-R test. If 'nChains_ConvergeTest' is zero, both values of 'convergeTest_startsFile' and 'outConvergenceTest' are empty. 'DAresultsPath' is the directory saving the results of DA whose directories are also listed in the following six lines: 'outJ' for the accepted mismatches; 'outC' for the accepted parameter values; 'outRecordNum' for the number of accepted parameter values; 'outBestSimu' for the best simulation outputs with the optimal parameter values; 'outBestC' for the optimal parameter values. For MIDA without GUI, 'display_plot' indicates whether or not to visualize the posterior distributions after DA.

[Figure]

**Figure C1**. An example of the 'namelist.txt' file. In order to use MIDA, users need to prepare data and a model and specify their file names and directories in the 'namelist.txt' file. "

*22. Line 263: Make sure to cite all software packages.*

Thanks for the kind reminder. We added citations of all Python packages used in MIDA in L289. The updated sentence is "For the non-GUI version, users need to install Python 3.7 and relevant packages (i.e., numpy, pandas, shutil, subprocess, matplotlib, math, os, and seaborn)."

*23. Line 314: Helpful examples. Really curious where 302 years of leaf area data come from!*

We first generated simulation results of the BiomeE model with forest ages ranging from 0 to 800 years, then compared the simulated forest height or vertical structure to GEDI-derived observations and determined the forest age is 302 yrs. According to the empirical knowledge, the forest is around 300~350 years old and a variation of 50 years is acceptable as because forests are almost equilibrated.

*24. Line 343: "complex reasons" is somewhat vague*

We are sorry about the vague description. We added specific descriptions "such as improper prior parameter range" for the reasons leading to edge-hitting posterior distribution in L364. In addition, we included other possible reasons to explain the results. The sentences are revised as "The edge-hitting posterior distributions result from complex reasons, such as improper prior parameter range. Users may change the prior ranges to examine if those posterior distributions can be improved or examine correlations among estimated parameters. "

*25. Line 499: Citation?*

Thanks for the suggestion. We added a citation of Gao et al. (2011) in L521.

*26. Line 504: Not sure about "first" unless you make your definition of model agnostic more specific*

We are sorry about the unclear statement. We changed the sentence to "Compared to the model-independent DA tools mentioned above, MIDA is the first that uses the MCMC method for DA.'" in L526. It echoes with L512-516.

*27. Figure 1: Very nice!*

Thanks.

*28. Figure 6: the Cis appear homogenous. Can MIRA deal with heteroskedasticity?*

Thanks for your question. MIDA uses MCMC algorithms which requires homogenous variance. MIDA is also flexible enough to incorporate other algorithms that can deal with data of heteroskedasticity.

*29. Figure 7: the colors in the legend have an extra character that should be removed*

As suggested, we removed the extra character in the legend of model no. in Fig. 7 as shown below.

[Figure]

*30. Figure 4, 5, and 8: these posterior distributions do not look converged to me. Could you also include the mcmc chains as well here or in the supplements?*

Thanks for your question. These cases study are to repeat published DA results to demonstrate the capability of MIDA. Taking Fig. 4 as an example, the constrained posterior distributions are similar to those from the original study in Lu et al. (2017). In addition, MIDA has incorporated algorithms to support Gelman-Robin convergence test of multiple MCMC chains as described in L354-359.